



# Extended range luminescence dating of quartz and alkali-feldspar from aeolian sediments in the eastern Mediterranean

Galina Faershtein[1,2], Naomi Porat[1], Ari Matmon[2]

[1] Geological Survey of Israel, 32 Yesha'ayahu Leibowitz St., Jerusalem 9692100, Israel
[2] Institute of Earth Sciences, The Hebrew University of Jerusalem, Jerusalem 91904, Israel

*Correspondence to*: Galina Faershtein (galaf@gsi.gov.il)

**Abstract.** Optically stimulated luminescence (OSL) on quartz is an established technique for dating late Pleistocene to late Holocene sediments. Unfortunately, this method is often limited to up to 100 ka (thousands of years). Recent developments in new extended range luminescence techniques show great potential for dating older sediments of middle and even early

Pleistocene age. These methods include thermally transferred OSL (TT-OSL) and violet stimulated luminescence (VSL) for quartz and post infrared-infrared stimulated luminescence (pIRIR) for feldspar. Here we investigate the luminescence behavior of the TT-OSL, VSL and pIRIR signals of quartz and feldspar minerals of aeolian sediments of Nilotic origin from the eastern Mediterranean. We sampled a 15 m thick sequence (Kerem Shalom) comprising sandy calcic paleosols, which is part of a sand sheet that covers an extensive region in south-western Israel. Dose recovery and bleaching experiments under

natural conditions indicated that the $pIRIR_{250}$ signal is the most suitable for dating the Nilotic feldspar. Luminescence intensity profiles revealed natural saturation of the three signals at the same depth of ~6 m, indicating that ages of samples below that depth are minimum ages. Using TT-OSL and $pIRIR_{250}$, a minimum age of 715 ka, for the base of the section was obtained, suggesting aeolian sand accumulation along the eastern Mediterranean coastal plain already since the early Pleistocene. Our results indicate that both TT-OSL and $pIRIR_{250}$ can accurately date middle Pleistocene aeolian sediments of

Nilotic origin and that minimum ages can be provided for early Pleistocene samples.

## 1 Introduction

Dating clastic sediments of Pleistocene age, particularly of middle and early Pleistocene, is an ongoing challenge. Several methods are available, but each has its limits. Magnetostratigraphy is binary (reverse or normal polarity) and has low resolution (extended periods with no reversals); Cosmogenic radionuclide (CRN) burial ages (Gosse and Phillips, 2001)

could suffer from unknown inherited ratios and complex post burial production which would result in under or over estimation of the ages and carry large uncertainties (e.g. Granger, 2006, Davis et al., 2012). U-Th and U-Pb isotopic systems are restricted to pure carbonates (not common in clastic environments) while the former is limited to ~500 ka; and Ar-Ar dating requires the presence of volcanoclastic deposits.



Luminescence dating, especially optically (blue) stimulated luminescence (OSL) on quartz, is an established and reliable
dating technique for terrestrial and shallow marine sediments of late Pleistocene to late Holocene time scale (Wintle and
Adamiec, 2017). The OSL method is especially essential in arid areas where there is a lack of organic material for $^{14}$C
dating. This method indicates the last exposure of the mineral (quartz of alkali-feldspar) grains in the sediments to sunlight.
The luminescence signal accumulates over time due to environmental ionization radiation, as electrons are trapped in defects
within the mineral lattice. The age is calculated from the ratio of the equivalent dose ($De$) to the environmental dose rate
($Dr$). The (blue) OSL is limited by the saturation of the luminescence signal, occurring at ~150 Gy in most cases (e.g.
Chapot et al., 2012).

Over the last decade several novel methods were proposed in order to extend the range of the luminescence dating into the
middle and even early Pleistocene. These include thermally transferred OSL (TT-OSL) and violet stimulated luminescence
(VSL) for quartz, and post infrared-infrared (pIRIR) stimulated luminescence at elevated temperatures (up to 290 °C) for
alkali-feldspars (Wang et al., 2006a; Jain, 2009; Thomsen et al., 2008). Initial results suggested potential for dating
sediments of up to 1 Ma age (Wang et al., 2006b; Ankjaergaard et al., 2013; Buylaert et al., 2012). Nevertheless, a more
comprehensive investigation revealed different limitations of using these signals. For example, the TT-OSL signal is
thermally unstable, therefore producing only minimum ages after a few hundred kyr (Adamiec et al., 2010; Shen et al., 2011;
Chapot et al., 2016; Faershtein et al., 2018); it appears that the natural growth of the VSL signal cannot be properly
described with single aliquot regenerative (SAR) constructed dose response curve (DRC) generally used for $De$
determination (Ankjaergaard et al., 2016; Ankjaergaard 2019); there is evidence of age overestimation for the pIRIR$_{290}$ and
athermal signal loss (termed anomalous fading) issues for the pIRIR signals measured at lower temperatures (Lowick et al.,
2012; Tsukamoto et al., 2017). The potential and limits of these methods in dating early and middle Pleistocene sediments
were tested in several locations around the globe (e.g. Zander and Hilgers, 2013; Arnold et al., 2015).

The eastern Mediterranean coastal plain is mostly underlain by Pliocene- marine and Pleistocene shallow marine and aeolian
sediments of Nilotic origin (Gvirtzman et al., 1984; Almagor et al., 2000; Crouvi et al., 2008; Amit et al., 2011; Muhs et al.,
2013), which are rich in quartz and contains smaller amounts of feldspar. Both minerals have excellent luminescence
properties and in the last twenty years have been extensively used for dating in this region (e.g. Porat et al., 1999, 2004,
2008). The youngest of these sediments, close to the Mediterranean coastline, have been comprehensively dated in the past
by the luminescence methods (quartz OSL and feldspar IRSL$_{50}$), mostly up to 70 ka (e.g. Porat et al., 2004; Mauz et al., 2013
and references within). Recently, extended range luminescence techniques (TT-OSL and pIRIR), as well as CRN burial
dating added new middle and early Pleistocene ages to the local chronology (e.g. Davis et al., 2012; Harel et al., 2017;
Shemer et al., 2018). The new data strongly suggest sediment accretion since the late Pliocene - early Pleistocene, associated
with westward shift of the coastline (Haler et al., 2017). In order to deepen our understanding of the sedimentological
evolution of the coastal plain we investigate the suitability of the extended range dating methods to date the local Nilotic
sediments.





A representative exposure of the Pleistocene aeolian sediments is located at the sand sheet of Kerem Shalom (KR), 13 km from the Gaza Strip coastline (Fig. 1). This is a 15 m thick section (exposed in a trench) composed of seven sandy calcic paleosols units, which has been described in detail by Zilberman et al. (2007). In brief, the units are (from the base): unit 1 –

friable sand with four amalgamated well developed Bk calcic horizons (stage III-IV); unit 2 – sand with well developed Bk calcic horizon (stage III-IV); unit 3 – sand with two calcic paleosols (stage III-IV); unit 4 – silty sand with clay horizon at the top; unit 5 – silty sand with stage III Bk calcic horizon at the top ; unit 6 – friable sand at the bottom and a paleosol with Bk calcic horizon at the top (stage II-III); unit 7 – friable sand with some carbonate nodules and pottery fragments at the top. The depositional unites are separated by sharp contacts and contain evidence of bioturbation such as burrows and rhyzolites.

This distinct sequence reflects a cyclic process, which starts with relatively rapid deposition of aeolian sand and continue with a long period of stability associated with the growth of vegetation, dust accumulation and soil development (Zilberman at al., 2007). The section was previously dated with OSL to between 480 ka and 13 ka (Zilberman et al., 2007); however, Faershtein et al., (2019) showed that the OSL ages should be considered as minimum ages for all samples below 2 m due to natural signal saturation.

It was recently demonstrated that the quartz from KR is thermally stable with excellent luminescence properties (Faershtein et al., 2018). Preliminary paleomagnetic measurements suggested reverse polarity at the base of the section (Ron, personal comment). Thus, the KR sediments allow us to test the extended range dating methods. The low environmental dose rates of the sand layers, ~0.5 Gy ka$^{-1}$ for quartz and ~1.0 Gy ka$^{-1}$ for k-feldspar, predict equivalent doses of 390 Gy and 780 Gy for quartz and feldspar, respectively, for the lowest sample (15.3 m). Theoretically the extended range methods could easily

measure such doses. Therefore, the KR section is a perfect sequence for testing the applicability of these methods for the eastern Mediterranean sediments originating from the Nile. This paper presents a comprehensive investigation of the luminescence behavior of TT-OSL, VSL, and pIRIR signals for these sediments. Bleaching and dose recovery experiment are performed; the section is dated with TT-OSL and pIRIR$_{250}$; and VSL multiple aliquot additive dose (MAAD) DRC is constructed. The reliability of the ages and their geological implications are discussed.

## 2 Methods

Sixteen samples were collected from the KR section by drilling horizontally holes, 30 cm into the sediment. After discarding the sediment from the outer 10 cm, the samples for chemical analysis and luminescence measurement were further treated. In addition, a modern sample was collected from the top bed in a nearby pit (KR-17).

Sample preparation and measurements were carried out under weak orange-red light. The separation procedure included wet

sieving to 74-105, 88-125 or 125-150 μm; dissolving carbonate with 8% HCl solution; magnetic separation using a LB-1 Frantz magnetic separator at a current of 1.4 A on the magnet (Porat et al., 2006); etching the non-magnetic fraction in concentrated 40% HF solution for 40 min, and additional soaking in 16% HCl overnight to dissolve any fluorides which may have precipitated (Porat et al., 2015). The alkali-feldspar was extracted from the non-magnetic fraction by density separation





to <2.58 gr cm$^{-3}$ with heavy liquid (Sodium-Polytungstate) and short etching for 10 min with 10% HF solution (Porat et al.,

2015; for details see supplementary material). Due to lack of material, feldspar was not extracted for sample KR-4.

Alpha, beta, and gamma dose rates were calculated from the concentration of the radionuclides U, Th, and K measured by ICP-MS (for U and Th) and ICP-OES (for K), with uncertainties of 5%, 10%, and 3%, respectively. Internal K content in the feldspars was estimated at 12.5±0.5% (Huntley and Baril, 1997). The *a*-value was estimated at 0.15 ± 0.05, an average of the values given for alkali-feldspar by Balescu et al. (2007) and Rendell et al. (1993). Gamma and cosmic dose rates were

measured in the field with a portable gamma counter. Water content was estimated at 5±2% as typical of sands in this arid region (Zilberman et al., 2007). The dose rates data is presented in Table 1.

All measurements were undertaken using TL/OSL DA-12 or DA-20 readers, equipped with blue LEDs, solid state violet (405 nm) laser diode, and IR diodes for stimulation. Irradiation was by calibrated $^{90}$Sr $\beta$ sources with dose rates of 0.04 or 0.97 Gy s$^{-1}$ respectively. Detection was through 7.5 mm U-340 filters for quartz and a combination of Schott BG-39 and

Corning 7-59 filter pack for feldspar. For TT-OSL and VSL, 5 mm aliquots on aluminum discs were used for measurements, unless stated otherwise. For feldspar 2 mm aliquots on stainless steel cups were used.

The SAR protocol was applied for *De* determination for the OSL, TT-OSL and pIRIR$_{225,250,290}$ (Murray and Wintle, 2000; Porat et al., 2009; Thiel et al., 2011). Measurement details are listed in Table 2. Average *De* values and errors were calculated using the central age model (CAM) after removing distinct outliers (Galbraith and Roberts, 2012).

Based on the bleaching and dose recovery experiments (Sect. 3.2.3 and 3.3.2), the 280 °C preheat temperature and 250 °C stimulation temperature were used for the pIRIR *De* measurements. Anomalous fading was assessed through fading experiments (as in Auclair et al., 2003) measured on three sensitized aliquots for most samples. IRSL response to a 100 Gy $\beta$ dose (normalized to a 30 Gy test dose response) was repeatedly measured after storage for 15 min and up to 48 to 84 hours. The *g*-value (% per decade), normalized for 2 days, and the recombination center density ($\rho$) were determined using the

analyse_FadingMeasurement R function (Kreutzer and Burow, 2019) following the IRSL luminescence decay model of Huntley (2006). The averages with standard divisions of the *g*-value and $\rho$ were further used for fading corrections. For samples KR-11 to KR-15, the fading rates were not measured and their *g*-value and $\rho$ were assessed from the nearest samples. Fading corrections of Huntley and Lamothe (2001) and Kars et al. (2008) were both applied to the final calculations. The Huntley and Lamothe (2001) correction was used on samples from the upper 6 m, as it is suitable only for

the linear part of the DRC. It was preformed using the *g*-value with the calc_FadingCorr R function (Kreutzer, 2019). The Kars et al. (2008) correction reconstructs a natural simulated DRC and projects the natural IRSL onto that DRC to produce the fading corrected age. The calc_Huntley2006 R function was used (King and Burow 2019). This function requires the laboratory DRC with *Ln/Tn* and the $\rho$ parameter for the simulated DRC construction. First, the calc_Huntley2006 was applied to all aliquots of samples KR-1, which were previously used for *De* determination. Then the function was applied

using the average *Ln/Tn* value (with standard deviation) and a combined DRC of these aliquots. As the average output parameters were almost identical (0-4 % difference), the average *Ln/Tn* and the combined DRCs were used for all other samples.





A DRC constructed by the SAR protocol, which is the most commonly used for *De* determination, fails to mimic the natural growth of the VSL signal (Ankjærgaard et al., 2016). This difference is attributed to sensitivity changes during preheat

which is applied prior to the violet stimulation in the measurement protocol (Table 2). On the other hand, a DRC constructed on a modern sample using a MAAD approach (Aitken, 1998) is much closer to the natural DRC (Ankjærgaard et al., 2016; Ankjærgaard, 2019). Adopting the MAAD approach, a MAAD DRC was constructed for the modern sand sample DF-13 with an OSL age of 40±10 years (Roskin et al., 2011a; Table S2). Forty-eight fresh aliquots were prepared and divided into 8 groups. Each group of aliquots was irradiated with increasing beta doses (0, 50, 100, 200, 400, 600, 800, 1000 Gy). The VSL

signal of the aliquots was then measured and normalized to the VSL signal of a 490 Gy test dose (Table 2), to construct a MAAD DRC (Fig. 2). The DRC can be fitted equally well with an exponential plus linear ($R^2$=0.997) and double exponential ($R^2$=0.999) functions. When fitted with the exponential plus linear function, the characteristic dose of the exponential component is $D_0$=69±22 Gy. For the double exponential function, the characteristic doses are $D_{0,1}$=43±28 Gy and $D_{0,2}$=369±322 Gy.

As the MAAD DRC is comparable to the natural DRC (Ankjærgaard et al., 2016), it is expected that DRCs constructed for different samples would be comparable as well. In order to explore this assumption as an alternative route for using the MAAD approach for VSL dating, the MAAD protocol was applied to sample RUH-180 from the Ruhama section, about 50 km to the north-east from KR (Fig. 2; Table S2). The TT-OSL *De* value of this sample is 163±15 Gy, corresponding to 126±5 ka, within the reliable dating range of the TT-OSL method (Faershtein et al., 2018); therefore, it was used as an age

control. The RUH-180 MAAD DRC was plotted with the addition of 160 Gy on each dose point on top of the DF-13 MAAD DRC (Fig. 2). It is clear that when assuming a *De* value of 160 Gy for RUH-180, the two MAAD DRCs overlap. It seems that comparison of a sample's MAAD DRC with the DRC of a modern sample is the right step toward developing the VSL dating method. Perhaps the sliding technique used for Infrared radiofluorescence (IR-RF) can also be used (Erfurt and Krbetschek, 2003; Frouine et al., 2017). This direction was not investigated further and is beyond the scope of this paper.

Most experiments were conducted on the KR samples. However, due to small sample size, some of the tests were performed on samples from other sites, on aeolian sediments also originating from the Nile. For additional information regarding these samples see supplementary material.

## 3 Results and discussion

### 3.1 Luminescence signals and dose response curves

Representative luminescence signals and DRCs of the KR samples are shown in Fig. 3, displaying good luminescence properties: For all samples, the OSL signal is dominated by the fast component; Recycling ratios are mostly within 5% of unity; and there is no significant feldspar contamination in the quartz grains as insured by the negligible IR depletion ratio (Duller, 2003). However, the *De* values of most samples are above ~150 Gy, which is considered the upper limit for OSL dating of Nilotic quartz (Faershtein et al., 2019; Table 3).



The TT-OSL signal is significantly dimmer than the OSL signal and the background level is 15-25% of the natural signal. The laboratory DRC grows linearly up to high doses (at least 600 Gy), with good recycling ratios, within 10%, for most measured aliquots. The VSL signal decays slowly to a background level which is ~10% of the natural signal. The natural VSL signal and a response to a 490 Gy test dose have a similar shape. No SAR DRCs were constructed for the VSL signal, as discussed in Sect. 2. The pIRIR$_{250}$ signal is bright and is reduces to 10% within 20 seconds. The recycling ratios are within

the acceptable 10% of unity and recuperation is smaller than 2% (except for the modern sample). The laboratory constructed DRC saturates at 700-800 Gy.  Average fading rate measured for the pIRIR$_{250}$ signal is 1.4±0.2% per decade.

### 3.2 Bleaching

Bleaching experiments were performed under natural sunlight, during the sunny and cloudless eastern Mediterranean summer. Freshly prepared aliquots were covered with a transparent Plexiglas and left outside at a spot which receives direct

sunlight for 8 h a day, for various time durations. Experiment details for each signal are listed in Table 4.

### 3.2.1 TT-OSL

Sample RUH-300 (Table S2), from the Ruhama site, was used for the experiment. This sample has OSL and TT-OSL *De* values of 214±11 Gy and 264±11 Gy, respectively. Early- and late-background signal subtractions were used for comparison to check for better separation of the bleachable component. Fig. 4a presents the bleaching experiment results. There is no

significant difference between the bleaching rates calculated using early and late backgrounds. The normalized TT-OSL signal decreased to 50% after ~4 h of exposure to direct sunlight. Further exposure to sunlight reduced the signal to 20% after 64 h (8 days) and to 11% after 148 h (18.5 days). These results are in agreement with those of Tsukamoto et al. (2008) and Porat et al. (2009). The relatively slow bleaching rate of the TT-OSL signal suggests that this signal is suitable for dating aeolian sediments that experience prolonged exposure to sunlight during transport prior to final sedimentation. Indeed, very

low TT-OSL *De* values of 2-4 Gy were measured on modern aeolian samples from the region (e.g. KR-17 and DF-13; Table S2). High residual doses of over 100 Gy were reported elsewhere for fluvial sediments (Hu et al., 2010; Duller et al., 2015), implying low suitability of the TT-OSL signal for dating such sediments. Nevertheless, samples of early-middle Pleistocene age from different sedimentation environments are in agreement with control ages (Arnold et al., 2015). Therefore, it seems that bleaching issues are not significant for dating samples in this time range.

### 3.2.2 VSL

The bleaching of the VSL signal was investigated using sample KR-10. It was chosen since it is considerably old but homogeneous based on TT-OSL *De* distribution (TT-OSL *De*=278±12 Gy; *OD*=18%). The results show that after 120 h of solar bleaching, the residual VSL signal is ~15% (Fig. 4b). Assuming that the VSL *De* should be similar to the TT-OSL *De* estimate of 280 Gy, these 15% correspond to ~42 Gy. Fitting the data suggests that 20 h of sunlight are required to reduce

the VSL signal by 50%. Previous studies reported on lower residuals signals of 6-35 Gy after bleaching in a solar simulator





(Ankjaergaard et al., 2013; Hernandez and Mercier, 2015). The natural signal *Ln/Tn* of a modern sample from the region, DF-13, was found to be 3.5% of the *Ln/Tn* of KR-10, corresponding to ~10 Gy, implying sufficient bleaching in nature under suitable conditions. Also, VSL ages in agreement with other luminescence ages were reported from the coastal plain of Israel (Porat et al., 2018). Therefore, it seems that bleaching in nature is adequate probably due to long exposure to sunlight

throughout the aeolian transport.

### 3.2.3 pIRIR

The bleaching of the $pIRIR_{225,250,290}$ signals was investigated using sample KR-8. The $IRSL_{50}$ (measured as part of the $pIRIR_{290}$) and $pIR\text{-}IR_{225,250,290}$ signals, measured after the different bleaching durations, are shown in Fig. 4c. The $IRSL_{50}$ signal droped to 1% after 4 h of exposure. The pIRIR signals are bleached to a lesser degree, yet all three signals were

bleached to less than 10% after 4 h and to less than to 2% after 64 h of exposure to direct sunlight. This implies a full signal resetting of the pIRIR signals at deposition for aeolian sediment.

### 3.3 Dose recovery

### 3.3.1 TT-OSL

Samples RUH-40 and RUH-90 were used for dose recovery experiment (Table S2). These are the two uppermost samples

from the Ruhama site with TT-OSL *De* values of 42±2 Gy and 53±3 Gy, respectively. Prior to the dose recovery measurements, fresh aliquots were bleached by sunlight for 10 and 18.5 days for RUH-90 and RUH-40, respectively (Table 4). Three doses were recovered; 200, 450, and 700 Gy. After a 10 h pause, the TT-OSL *De* was measured using the SAR protocol (Table 2). Early and late background subtractions were used for comparison.

Using late background subtraction for the TT-OSL signal yielded a much better recovery than early background subtraction

(Fig. 5a); the latter overestimated the given doses by 16-77%. Using late background, the 450 Gy given dose was perfectly recovered. For the other two doses, 200 and 700 Gy, the late background subtraction resulted in overestimation of 32-37% and 8%, respectively. The recovery ratios of the 200 Gy dose are almost identical for the two samples, 1.32 and 1.37. In order to check whether there is a significant residual dose, which might affect the recovered dose, the *De* values of two additional aliquots of RUH-40, bleached for 18.5 days, were measured. It appears that a small residual dose of 6-7 Gy still

remains after the prolonged sun bleaching, however this is only 1.5-3.5% of the given dose in our experiment and cannot explain the substantial overestimation for the 200 Gy recovery. Porat et al. (2009) carried out a dose recovery experiment on a modern sample from KR (KR-17). They achieved a better recovery for the 700 Gy dose, which might be explained by slightly different measurement conditions. In both experiments there is some overestimation at the lower doses, which is less significant for the high doses that TT-OSL is usually used for measuring.





### 3.3.2 pIRIR

A modern coastal sample was used for this experiment (ML-D-13; Table S2). Beta doses of 100, 400, and 900 Gy were given and recovered after a pause of 25-48 h. For all the recovered doses a test dose of 30 Gy was used. The pIRIR$_{225,250}$ signals show excellent recovery of 97-102% for the three given doses with good recycling ratios (Fig. 5b). PIRIR$_{290}$ results show some overestimation at 400 Gy and significant overestimation at 900 Gy, 120% of the given dose. Fading measurements of the three pIRIR signals indicated low $g$-values of < 1.6% per decade for the three signals. Overall, the pIRIR$_{250}$ signal displays a preferable balance between bleaching time and the ability to recover a known dose. Thus, it was further used for $De$ determination.

### 3.4 Natural saturation profiles

In long, continuous profiles, natural saturation of the luminescence signals can be observed by plotting the natural signals of samples against their depth (Liu et al., 2016). Faershtein et al. (2019) constructed such profiles for the KR section using the OSL and TT-OSL signals (Fig. 6). Now we added the natural saturation profiles for the VSL and pIRIR$_{250}$ signals. Natural signals (normalized to the corresponding test dose) of 4 aliquots were measured for each sample (Table 2) and plotted against sample's depth (Fig. 6).

It was shown by Faershtein et al. (2019) that the natural OSL signal at the KR section increases for samples up to 2 m depth and from there downwards it is constant. As the section is composed of seven superimposed well developed calcic paleosols, each requiring prolonged time to develop, rapid sedimentation of the lower 13 m is not likely. Rather, the natural OSL signal of these samples has stopped growing over time and is saturated. The saturation depth of the OSL signal emphasizes that the OSL ages reported by Zilberman et al. (2007) are minimum ages (except for the upper 3 samples). Similarly, the natural TT-OSL (Faershtein et al. 2019), VSL and pIRIR$_{250}$ signals grow to a depth greater than the OSL – up to 6 m, however they are constant for deeper samples (Fig. 6). There are four paleosols below that depth, therefore it is unlikely that all deeper samples are of the same age, implying field signal saturation.

Remarkably, the three signals (TT-OSL, VSL and pIRIR$_{250}$) reach their maximum luminescence at the same depth. One explanation could be that they reach natural saturation at the same dose. To explore this option, we examine the natural saturations of these three signals at the Luochuan loess section in China, where natural DRCs were constructed (Chapot et al., 2016; Ankjærgaard et al., 2016, Li et al., 2018). There, natural DRCs suggest field saturation at about 2000 Gy for both TT-OSL and VSL. Seemingly, that data supports the similar saturation dose at KR. However, the two signals have different thermal stabilities. Faershtein et al. (2018) showed that for sediments with different environmental dose rates, the thermally unstable TT-OSL signal reaches saturation at different doses. Indeed, for the KR sediments (average dose rate of 1.2±0.3 Gy ka$^{-1}$), the natural TT-OSL signal saturates at ~500 Gy (Faershtein et al., 2019), a much lower dose than at Luochuan. Regarding the thermal stability of the VSL source trap, Ankjærgaard et al. (2013) reported a lifetime of $10^{11}$ years (at 10 °C), implying that the natural saturation dose should not be affected by the sediment's dose rate; so, it is expected to be at a



comparable dose everywhere. For the pIRIR signal (stimulated at 225 °C) the natural DRC at Luochuan reaches the $2D_0$ (85% of saturation; Wintle and Murray, 2006) threshold at ~900 Gy, a much lower dose than the TT-OSL and VSL signals. To conclude, it is not likely that the three signals would reach natural saturation at the same dose at the KR section.

An alternative explanation for multiple signals reaching saturation at 6 m depth is a significant hiatus in sedimentation, whereby the sediments below 6 m are much older than those above 6 m. There are field evidences supporting this option: Soil unit 5, below the saturation depth, has a highly developed calcic Bk horizon (stage III) which requires tens of thousands of years to form (Birkeland, 1999); the unit has a higher clay and silt content compared to the other paleosols, suggesting long surface exposure with clay enrichment of the sand, also requiring tens of thousands of years (Gile et al., 1966; Danin 260 and Yaalon, 1982). Field saturation at 6 m indicates that accurate dating can be provided only for the upper part of the section.

Another way to assess the evolution of the natural OSL, TT-OSL, and pIRIR signals is to construct a semi-natural DRC, by plotting the natural signals against the laboratory measured *De* values, as was demonstrated for the OSL and TT-OSL signals at KR (Faershtein et al., 2019). The three signals display a common behavior; the natural signal grows with measured *De* up 265 to a certain value and then stays constant, indicating that in the laboratory signals grow beyond natural saturation (Fig. 7). The OSL *Ln/Tn* reaches its maximum value at relatively low dose of about 100 Gy. When the KR data is combined with many other sites with quartz of Nilotic origin, it is evident that the natural OSL reaches the $2D_0$ limit at ~140 Gy (Faershtein et al., 2019), somewhat higher than the KR section when plotted alone. This suggests that when possible, a multi-sites comparison is needed for regional characteristics of the luminescence behavior.

The TT-OSL *Ln/Tn* grows to about 400 Gy and is constant for higher doses up to 500 Gy, beyond which there are no *De* values (Fig. 7b). The growth of the TT-OSL signal in nature is limited by the low thermal stability of its traps. The lifetime of its main source trap under the environmental conditions at KR was calculated to about 550 ka using both field and laboratory data (Faershtein et al., 2018, 2019); this low lifetime explains the absence of higher *De* values. Closer examination of the saturated samples reveals that for samples with higher environmental dose rate the *De* is higher and the 275 TT-OSL ages are younger, as expected from the model simulations of Faershtein et al. (2018; Table 3).

Regarding the pIRIR$_{250}$, it seems that the natural signal grows up to 260 Gy and is constant for higher *De* values. However, the non-saturated sample KR-10 has a higher *De* value of $382\pm15$ Gy (Fig. 7c). This suggests that, perhaps, samples KR-11 to KR-13 are outliers with saturated pIRIR$_{250}$ signals and relatively low *De* values of 260-290 Gy. In that case, the natural saturation level is reached at 450 Gy; which is still low compared to the saturation level of the natural DRC constructed for 280 the Luochuan section in China (for pIRIR$_{225}$; Li et al., 2018). The natural signal growth is limited by the anomalous fading (Wintle et al., 1973; Thomsen et al., 2008). The *g*-values of the KR samples range between 1.2-1.7 % per decade, which are considered low and usually do not require correction (Buylaert et al., 2012). Nevertheless, fading rates increase over geological time (Wallinga et al., 2007) and should be corrected for (Li et al., 2019). Field saturation of the pIRIR signals is expected when equilibrium between trap filling due to ionizing radiation and electron escape through tunneling is achieved 285 (Huntly and Lian, 2006). This is expected to happen at lower doses than the laboratory saturation dose (Li et al., 2018).




There are no other published pIRIR$_{250}$ ages from the area; therefore, it is not clear whether the relatively low limit of 450 Gy is characteristic of the local feldspar or it is site dependent. It is possible that pIRIR signals stimulated at different temperatures have different saturation levels. PIRIR$_{290}$ ages (corresponding to $De$ values as high as 1600 Gy) in agreement with expected ages were reported elsewhere (Buylaert et al., 2012; Thiel et al., 2012; Zander and Hilgers, 2013).

Overall, inspection of the natural signals can be very informative and increase our confidence in distinguishing between reliable ages below saturation limit and samples that are already saturated. Construction of natural saturation profiles, as demonstrated here, can reveal saturated samples and treat them accordingly.

### 3.5 TT-OSL and pIRIR$_{250}$ ages

The TT-OSL ages range between 3.0±0.3 ka for the modern sample to 624±63 ka at a depth of 12.5 m (Table 3). The ages
are in stratigraphic order excluding one reversal at 8 m depth. The natural saturation profile revealed constant $Ln/Tn$ for the lower part of the section with clustered $De$ values of 400-500 Gy (Figs. 6, 7); yet, the ages increase with depth (Fig.8). This can be explained by the decrease in environmental dose rate with depth (Table 1). The TT-OSL ages below 6 m mirror the changes in dose rates with depth (Fig. 8).

The uncorrected pIRIR$_{250}$ ages range between 0.23±0.02 ka for the modern sample to 647±63 ka for the lowermost sample
(15.3 m; Table 5). The ages increase with depth, although there are two reversals at 5 and 11 m depth. There is a good agreement between the TT-OSL and the uncorrected pIRIR$_{250}$ ages up to 6 m depth (except for samples KR-5,9 at 4.1 m depth), where the signals reach their maximum $Ln/Tn$. At this depth ages of ~200 ka are obtained. From 6 m downwards the TT-OSL ages are mostly older that the pIRIR$_{250}$ ages. The ages converge again for the lowermost four samples at depths of 11-15 m.

The fading correction of Huntley and Lamothe (2001) was applied for samples from the upper 6 m (Table 5). The $g$-values vary between 1.17±0.36 to 1.66±0.28 % per decade, increasing the ages by 9-18%. The fading-corrected pIRIR$_{250}$ ages are between 0.25±0.02 ka and 254±15 ka. The Kars et al. (2008) correction was applied to all samples. The $\rho$ values range between 1.26±0.19*10$^{-6}$ and 1.70±0.34*10$^{-6}$. For most samples the simulated $D_0$ agree within 10% with the laboratory measured $D_0$ values. The fading-corrected pIRIR$_{250}$ aged range between 0.27±0.06 ka and 323±60 ka, with 17-72%
correction. The fading-corrected pIRIR$_{250}$ ages after Kars et al. (2008) tend to be higher the TT-OSL ages up to 6 m. For the samples at 6-11 m depth, for which the pIRIR$_{250}$ ages are younger than the TT-OSL ages, the fading correction does not compensate for the age difference. For the lower 5 samples the $Ln/Tn$ are above the saturation level of the natural simulated DRC (Fig. 9). Their fading corrected ages were determined to be older than the natural simulated 2$D_0$, up to >715 Gy.

The final chronology of the entire KR section was constructed as follows: For the upper 6 m, uncorrected pIRIR$_{250}$ ages were
used, as they are in excellent agreement with the TT-OSL ages. It is feasible that for the KR samples no fading correction is needed for samples younger than the field saturation level. For samples below 6 m the two signals are in field saturation as was indicated by the natural saturation profiles (Fig. 6). Thus, the ages are minimum ages. As the fading rates increase with time (Wallinga et al., 2007), pIRIR$_{250}$ corrected ages after Kars et al. (2008) were used for these field saturated feldspar





samples. As each method is limited by a different factor (thermal and athermal signal lose), there is no reason to prefer one
method over another; hence, the older age is considered as minimum age of the samples. The combined ages are in
stratigraphic order (Fig. 10), except for one reversal at 9.5 m depth (KR-14). Since the reversal is among minimum ages,
using the principle of super position, sample KR-14 is at least as old as sample KR-13 above it. So, the age of KR-14 is
considered to be >488 ka, similar to KR-13. Duplicate samples at 1.5 m and 4.1 m depths (KR-6,16 and KR-5,9 respectively)
have similar TT-OSL and pIRIR$_{250}$ ages, confirming the reproducibility of the two signals.

**3.6 VSL MAAD ages**

Ankjærgaard et al. (2016) suggested interpolating the natural VSL signals of samples on a MAAD DRC, of a modern
sample, in order to obtain their $De$ values. Following this approach, the natural signals of the KR samples were projected
onto the MAAD DRC (of DF-13) fitted with the exponential plus linear and double exponential functions. The resulting $De$
values were farther translated into ages using the samples' dose rates (Fig. 11). For the exponential plus linear fit, the errors
on the $De$ values and subsequently, on the ages, are 20-110% (Table S3). The large errors may be attributed to the low slope
of the linear component and the relatively large errors on the $Ln/Tn$ resulting from the weak signal. $De$ values obtained with
the double exponential function are slightly different (up to 15%) from those obtained with the exponential plus linear
function, with even larger errors (up to 500%; Table S3). The VSL ages obtained by the exponential plus linear function
were farther used for comparison with the other luminescence ages (Fig. 11). These ages are slightly lower than the TT-OSL
and the pIRIR$_{250}$ ages for the upper 6 m of the section. The ages of the lower samples are inconclusive due to the large
errors.

**3.7 Geological implications**

The KR outcrop presents a unique glimpse into the Pleistocene subsurface in the surrounding flat landscape. The sequence is
nearly complete: Although the contacts between the depositional units are sharp, the soil profiles are missing only their
uppermost part (A and upper B horizons), implying minor erosion, probably due to deflation (Zilberman et al., 2007). Cyclic
deposition was proposed, whereby sand deposition is followed by a stable period during which the calcic paleosols
developed, followed with minor erosion by deflation (Zilberman et al., 2007). This scenario is now refined, based on the new
and improved chronology.

The ages for the lower two thirds of the section (units 1-5) are not accurate as the units are too old for precise luminescence
dating. However, important information can still be deduced from the well-dated units 6 and 7. Unit 6 is 3 m thick and was
deposited during 80 ka (70-150 ka) in an average rate of 4 cm ka$^{-1}$ through a glacial and interglacial cycle (MIS 6-4). Thus, a
straightforward correlation between deposition of the KR sequence and Pleistocene climatic cycles cannot be made. The
stable period, in which the stage II-III paleosol of unit 6 was developed, continued for at least 55 ka (70-14 ka), the time
difference between the deposition of units 6 and 7. The soils that cap the underlying units (1-5) are more mature (stage III-
IV), implying longer stable periods between the earlier depositional cycles. Faershtein et al. (2018) demonstrated that the



evolution of the TT-OSL apparent age with time results in increasing age underestimation. Thus, it can be reasonably assumed that the time intervals between the minimum ages of the units represent the minimal time periods between their deposition. It appears that at least ~60-100 ka (differences in the minimum ages of the paleosols units) separate between each two depositional cycles (Fig. 10). This is in agreement with previous studies, which suggest that development of III-IV stage
calcic soil can take tens of thousands of years (Gile et al., 1981; Birkeland, 1999). When these gaps between units are summed up, the total time required for the deposition of the 7 units can be 800 ka.

When surfaces are stable, bioturbation is active, resulting in significant mixing that brings grains to the surface where their luminescence signal is reset, and inserts bleached grains tens of centimeters below the surface (e.g. Bateman et al., 2007). Thus, one can expect the A and upper B horizons to be kept relatively bleached all the time. Assuming rapid deposition of
the sand in each sandy paleosols units (Zilberman et al., 2007), this mixing can explain the relatively young age of sample KR-10 at the top of unit 5 (204 ka). While the rest of the samples from this unit are saturated with respect to TT-OSL and pIRIR$_{250}$ signals, sample KR-10 is only close to saturation. If the stable period between deposition of units 5 and 6 is as long as 100 ka, bioturbation can cause the significant age underestimation of the upper part of the unit. This phenomenon is not observed in unit 3 but can be observed in unit 2, where the minimum age obtained for sample KR-15 which was collected
from the upper part of the unit is 100 ka younger than the minimum age of sample KR-2, collected from the lower part of the unit.

Overall, it is suggested that units 4 and 5 were deposited >300 ka ago, unit 3 >500 ka, unit 2 >600 ka and unit 1 >700 ka. This implies that the accumulation of KR sand sheet has begun already in the early Pleistocene. The reversed polarity measured for unit 1 (Ron, personal comment) supports the early Pleistocene onset of the KR sequence.

The KR sand sheet is located at the boundary between two aeolian provinces: the Negev dune fields to the south and the coastal plain to the north. Aeolian sediments have been transported to the region by winds generally blowing from the west at least since the middle Pleistocene (Enzel et al., 2008, 2010; Roskin et al., 2011b). The extensive Negev dune field, which was stabilized after 18 ka (Roskin et al., 2011a), overlies late Middle to Late Pleistocene paleosols dated to 100-200 ka (Roskin et al., 2013). The absence of sediments dated to between 18 ka and 100 ka was explained by long-term aeolian
landscape equilibrium rather than erosion. During the stabilization after 18 ka, dunes over 10 m high were generated only 7 km south of KR (Roskin et al., 2011a). At the same time, at KR unit 7 which is only 1.5 m thick was deposited, indicating that despite the proximity, the KR section is different from the dune field province. In fact, the KR sediments are chronologically more comparable to the coastal plain eolian province (Zilberman et al., 2007). For example, during the deposition of unit 6 at KR, contemporaneous Kurkar ridges (aeolianite) were deposited along the coastal plain in several
pulses at 50-150 ka (Frechen et al., 2002, 2004; Porat et al., 2004; Sivan and Porat, 2004, Harel et al., 2017). Later, during the stable period between the deposition of units 6 and 7 (14-70 ka), the Natanya Hamra soil was developed along the coastal plain (13-57 ka; Porat et al., 2004; Shtienberg et al., 2017), also representing a stable period. The Kurkars and Hamra were dated mostly with IRSL$_{50}$, therefore their ages are most likely underestimated. In general, the accumulation rates at KR are low compared to the main aeolian province, perhaps due to the distance from the main sand source on the coast.





Calcic soils, similar to the KR paleosols, usually developed in semi-arid climate with annual rainfall of at least 200-250 mm per year (Birkeland, 1999), but some calcic precipitation can also be found in drier areas in sandy sediments (Amit and Harrison, 1995). According to Zilberman et al. (2007), the KR paleosols represent two climatic phases: a drier and windy climate during which the sand was delivered from the coast and accumulated, and a second, less windy and more humid climate in which vegetation was present on top of the sands, enabling dust trapping and soil development. This hypothesis

goes along with increased rain precipitation recorded by speleothem growth at 150-200 ka and 13-85 ka (Vaks et al., 2006). The speleothem record also suggests increased precipitation at 123-137 ka, during deposition of unit 6.

Wind velocity could have controlled the grain size of supplied sediment. It is suggested that during the windy and drier phase, sand was supplied to the area, while during the less windy phase, silt was supplied to the site as dust. Zilberman et al. (2007) dated two grain-size fractions from sample KR-7 (top of unit 6) to 42 ka and 55 ka for 74-105 μm and 150-177 μm,

respectively. They attributed the age difference to a later penetration of the silt into to the sandy soil. This sample was collected from a depth of 2.3 m, therefore is probably saturated with respect to the OSL signal. In the current study, only the 74-105 μm fraction of the sample was dated by TT-OSL and pIRIR$_{250}$, to 68±6 and 77±6 ka, respectively. Hence, the age difference between the two grain sizes cannot be verified. Silty eolian sediments from the northern Negev, known as primary loess, were dated by OSL mostly to 11-70 ka (Crouvi et al., 2008). These ages correspond to *De* values of 23-127 Gy, within

the reliable range for the Nilotic quartz (Faershtein et al., 2019). These ages are consistent with silty dust supply during the stable period between the deposition of the KR units 6 and 7.

## 4 Conclusions

A comprehensive investigation of the luminescence behavior of quartz TT-OSL, VSL, and feldspar pIRIR$_{225,250,290}$ signals of eastern Mediterranean sediments of Nilotic origin, was conducted using samples from the KR section. Bleaching

experiments under direct sunlight showed relatively rapid bleaching for the pIRIR signals and slower bleaching rates for the TT-OSL and VSL signals, suggesting that these two signals should be used for dating mainly aeolian sediments. Nevertheless, on a timescale of early-middle Pleistocene, sediment from other sedimentological environments can be dated with TT-OSL. Dose recovery experiments showed adequate recovery for TT-OSL and indicated that the pIRIR signal measured at 250 °C is the most suitable for dating the local sediments. Natural saturation profiles indicated that the natural

TT-OSL, VSL and pIRIR$_{250}$ signals of all samples deeper than 6 m are saturated. Therefore, the TT-OSL and pIRIR$_{250}$ signals used for dating of these samples provide minimum ages. Construction of such profiles is recommended on a local and a regional scale in order to reveal saturated samples. Comparison between TT-OSL and pIRIR$_{250}$ ages indicate that no fading correction is needed for the pIRIR$_{250}$ ages below natural saturation. Our results indicate that accurate ages can be provided for geological and prehistoric samples of middle Pleistocene age.

The multiple signal luminescence dating extended the dating range of the KR section into the early Pleistocene. Minimum ages of the lower units indicate that stable periods of soil development between each sand sedimentation cycle lasted for at



least 60 ka. The chronology of the KR section associates it mainly to the coastal plain sedimentological provinces, which sedimentary sequence is probably older than was previously though.

## Data Availability

The data can be received by communicating the corresponding author.

## Author Contributions

GF conducted the study and prepared the manuscript with input from all co-authors. NP and AM supervised and assisted GF through the study.

## Competing interests

The authors declare that they have no conflict of interest.

## Acknowledgements

This research was supported by the Israel Ministry of Energy (grant No. 214-17-005), and by THE ISRAEL SCIENCE FOUNDATION (grant No. 1871/16). We thank E. Zilberman for fruitful discussions and feedback during the work on this manuscript.

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





**Table 1: Dose rate data for the KR samples. Internal K content in the feldspars was estimated at 12.5±0.5% (Huntley and Baril, 1997, giving an internal dose rate of 372±63 µGy a⁻¹). Water content was estimated at 5±2% (Zilberman et al., 2007). Uncertainties on K, U and Th contents are 3%, 5% and 10%, respectively. Gamma and cosmic dose rates were measured in the field with a portable gamma counter.**

| Sample name | Unit | Depth (m) | Grain size (µm) | K (%) | U (ppm) | Th (ppm) | Ext. $\alpha$ (µGy a⁻¹) | Ext. $\beta$ (µGy a⁻¹) | Ext. $\gamma$+cosmic (µGy a⁻¹) | Quartz $Dr$ (µGy a⁻¹) | Feldspar $Dr$ (µGy a⁻¹) |
|---|---|---|---|---|---|---|---|---|---|---|---|
| KR-17 | 7 | 0.5 | 88-125 | 0.73 | 0.8 | 2.4 | 4 | 641 | 597 | 1241±65 | 1597±91 |
| KR-6 | 7 | 1.5 | 125-150 | 0.73 | 0.68 | 2.15 | 3 | 611 | 604 | 1218±65 | 1693±80 |
| KR-16 | 7 | 1.5 | 88-125 | 0.75 | 0.7 | 2.2 | 3 | 637 | 560 | 1200±62 | 1555±88 |
| KR-7 | 6 | 2.3 | 74-105 | 0.83 | 1.8 | 4.2 | 9 | 880 | 709 | 1598±77 | 1899±93 |
| KR-8 | 6 | 3 | 88-125 | 0.77 | 2.3 | 2.6 | 8 | 858 | 690 | 1556±75 | 1914±98 |
| KR-5 | 6 | 4.1 | 125-150 | 0.73 | 0.88 | 2.01 | 3 | 632 | 684 | 1319±73 | 1795±87 |
| KR-9 | 6 | 4.1 | 88-125 | 0.68 | 0.9 | 2.5 | 4 | 622 | 606 | 1232±66 | 1636±93 |
| KR-10 | 5 | 5.2 | 75-105 | 0.59 | 1.5 | 3.5 | 8 | 665 | 690 | 1363±74 | 1778±94 |
| KR-11 | 5 | 5.8 | 88-125 | 0.7 | 1.8 | 4.4 | 8 | 791 | 739 | 1538±79 | 1896±101 |
| KR-4 | 5 | 6.3 | 125-150 | 0.76 | 1.73 | 3.99 | 6 | 801 | 753 | 1560±80 | - |
| KR-12 | 4 | 7.2 | 88-125 | 0.73 | 2.2 | 4.5 | 9 | 863 | 826 | 1698±88 | 2056±108 |
| KR-13 | 3 | 8.2 | 88-125 | 0.38 | 1.7 | 2.8 | 7 | 529 | 478 | 1014±48 | 1371±80 |
| KR-14 | 3 | 9.5 | 88-125 | 0.45 | 1.4 | 2.4 | 5 | 529 | 739 | 1273±78 | 1630±100 |
| KR-3 | 3 | 10.7 | 125-150 | 0.38 | 1.3 | 2.64 | 4 | 469 | 579 | 1052±62 | 1528±78 |
| KR-15 | 2 | 11.7 | 88-125 | 0.29 | 0.9 | 1.7 | 4 | 344 | 382 | 730±44 | 1085±77 |
| KR-2 | 2 | 12.5 | 125-150 | 0.29 | 0.62 | 1.28 | 2 | 295 | 383 | 680±44 | 1086±77 |
| KR-1 | 1 | 15.3 | 125-150 | 0.27 | 0.56 | 1.49 | 2 | 280 | 398 | 680±45 | 1057±77 |





**Table 2: Measurement protocols and details of quartz TT-OSL and VSL and feldspar pIRIR signals used in this study.**


| | TT-OSL | VSL | pIR-IR$_{225/250/290}$ |
|---|---|---|---|
| **Discs/cups** | Aluminum discs | Aluminum discs | Stainless steel cups |
| **Aliquot size** | 5 mm | 5 mm | 1-2 mm |
| **Signal & background** | First 1 s & last 5 s | First 3 s & last 30 s | First 1 s & last 10 s |
| **Step** | | | |
| **1** | $\beta$ dose | $\beta$ dose | $\beta$ dose |
| **2** | TL at 260 °C for 10 s | TL at 300 °C for 100 s | TL at 255/280/320 °C for 60 s |
| **3** | Blue stimulation at 125 °C for 300 s | Blue stimulation at 125 °C for 100 s | IR stimulation at 50 °C for 200 s |
| **4** | TL at 260 °C for 10 s | | |
| **5** | Blue stimulation at 125 °C for 100 s (*Lx*) | VSL at 30 °C for 500 s (*Lx*) | IR stimulation at 225/250/290 °C for 200 s (*Lx*) |
| **6** | Test dose (2.2 Gy) | Test dose (490 Gy) | Test dose (30 Gy) |
| **7** | TL at 220 °C for 10 s | TL at 290 °C for 100 s | TL at 255/280/320 °C for 60 s |
| **8** | | Blue stimulation at 125 °C for 100 s | IR stimulation at 50 °C for 200 s |
| **9** | Blue stimulation at 125 °C for 100 s (*Tx*) | VSL at 30 °C for 500 s (*Tx*) | IR stimulation at 225/250/290 °C for 200 s (*Tx*) |
| **10** | Heat at 350 °C for 100 s | VSL at 380 °C for 200 s | IR stimulation at 350 °C for 300 s |



**Table 3: TT-OSL dating results of the Kerem Shalom samples. No. aliquots – number of aliquots used for *De* determination out of those measured. Average *De* values and errors were calculated using the CAM after removing distinct outliers (Galbraith and Roberts., 2012). OSL *De* and ages are from Zilberman et al. (2007) for comparison.**


| Sample name | Unit | Depth (m) | Quartz *Dr* (µGy a⁻¹) | OSL *De* (Gy) | Age (ka) | TT-OSL No. aliquots | *OD* (%) | *De* (Gy) | Age (ka) |
|---|---|---|---|---|---|---|---|---|---|
| KR-17 | modern | 0.5 | 1241±65 | 0.17±0.03 | 0.13±0.02 | 8/10 | 8 | 3.7±0.3 | 3.0±0.3 |
| KR-6 | 7 | 1.5 | 1218±65 | 16±2 | 13±2 | 10/10 | 0 | 17±1 | 14±1 |
| KR-16 | 7 | 1.5 | 1200±62 | 17±3 | 15±2 | 10/10 | 12 | 20±1 | 17±1 |
| KR-7 | 6 | 2.3 | 1598±77 | 66±6 | 42±5 | 10/10 | 22 | 108±8 | 68±6 |
| KR-8 | 6 | 3.0 | 1556±75 | 117±21 | 75±14 | 9/10 | 16 | 149±9 | 96±7 |
| KR-5 | 6 | 4.1 | 1319±73 | 123±15 | 93±13 | 10/10 | 10 | 205±13 | 155±13 |
| KR-9 | 6 | 4.1 | 1232±66 | 105±6 | 86±6 | 10/10 | 12 | 186±8 | 151±10 |
| KR-10 | 5 | 5.2 | 1363±74 | 203±46 | 149±35 | 9/10 | 18 | 278±12 | 204±14 |
| KR-11 | 5 | 5.8 | 1538±79 | 301±32 | 196±23 | 9/10 | 9 | 472±16 | 307±19 |
| KR-4 | 5 | 6.3 | 1560±80 | 240±36 | 154±24 | 10/10 | 25 | 477±37 | 306±29 |
| KR-12 | 4 | 7.2 | 1698±88 | 311±39 | 183±25 | 9/10 | 17 | 512±21 | 301±20 |
| KR-13 | 3 | 8.2 | 1014±48 | 333±67 | 326±68 | 10/10 | 15 | 495±25 | 488±29 |
| KR-14 | 3 | 9.5 | 1273±78 | 373±106 | 293±85 | 9/10 | 13 | 382±18 | 300±23 |
| KR-3 | 3 | 10.7 | 1052±62 | 246±21 | 234±25 | 10/10 | 18 | 499±29 | 475±39 |
| KR-15 | 2 | 11.7 | 730±44 | 266±77 | 364±108 | 9/10 | 13 | 406±18 | 555±42 |
| KR-2 | 2 | 12.5 | 680±44 | 325±58 | 478±91 | 10/10 | 24 | 424±33 | 624±63 |
| KR-1 | 1 | 15.3 | 680±45 | 287±58 | 422±91 | 10/10 | 18 | 373±22 | 549±47 |





**Table 4: Bleaching and dose recovery experimental details. For samples details see Table S2.**

**No. aliquots – number of aliquots measured for each experimental condition.**

|  | **TT-OSL** | **VSL** | **pIR-IR$_{225/250/290}$** |
|---|---|---|---|
| **Bleaching** |  |  |  |
| Sample used | RUH-300 | KR-10 | KR-8 |
| No. aliquots | 3 | 4 | 3 |
| aliquot size (mm) | 9 | 5 | 2 |
| Bleaching durations (h) | 0, 4, 8 16, 64, 148 | 0, 40, 120 | 0, 4, 8, 16, 32, 64 |
| Residual signal (%) | 11 | 15 | <2 |
| Residual $De$ (Gy) | 29 | 42* | <4 |
|  |  |  |  |
| **Dose recovery** |  |  |  |
| Sample used | RUH-40, 90 |  | ML-D-13 |
| No. aliquots | 3-4 |  | 4 |
| aliquot size | 9 |  | 2 |
| Bleaching duration prior to dosing (days) | 10-18.5 |  | 5 |
| Given doses (Gy) | 200,450,700 |  | 100, 400, 900 |
| Pause before recovery (h) | 10 |  | 25-48 |
| Recovery (%) | 100-137** |  | 97-100, 100-102, 100-121*** |

\* Assuming that the VSL $De$ is similar to the TT-OSL $De$ estimate of 280 Gy.

\*\* Using late subtraction.

\*\*\* For pIRIR$_{225,250,290}$, respectively.



**Table 5: PIRIR$_{250}$ dating results of the Kerem Shalom. Uncorrected and fading-corrected ages are presented.**

| Sample name | Unit | Depth (m) | Feldspar Dr (µGy a$^{-1}$) | g-value (% per decade) | $\rho$ (*10$^{-6}$) | No. aliquots | OD (%) | Measured $D_0$ (Gy) | Measured De (Gy) | Simulated $D_0$ (Gy) | Simulated De (Gy) | Uncorrected Age (ka) | Corrected Age[a] (ka) | Corrected Age[b] (ka) |
|---|---|---|---|---|---|---|---|---|---|---|---|---|---|---|
| KR-17 | modern | 0.5 | 1597±91 | 1.43±1.63 | 1.50±0.69 | 8/8 | 12 | - | 0.36±0.02 | - | 0.42±0.1 | 0.23±0.02 | 0.25±0.02 | 0.27±0.06 |
| KR-6 | 7 | 1.5 | 1693±80 | 1.47±0.18 | 1.55±0.20 | 8/8 | 17 | 303±1 | 25±1 | 289±1 | 34±6 | 15±1 | 17±1 | 20±4 |
| KR-16 | 7 | 1.5 | 1555±88 | 1.24±0.44 | 1.29±0.48 | 6/7 | 17 | 370±1 | 20±1 | 311±6 | 26±5 | 13±1 | 14±1 | 17±3 |
| KR-7 | 6 | 2.3 | 1899±93 | 1.49±1.09 | 1.58±1.16 | 8/8 | 19 | 256±10 | 145±10 | 256±4 | 216±46 | 77±6 | 89±13 | 114±25 |
| KR-8 | 6 | 3.0 | 1914±98 | 1.35±0.28 | 1.41±0.28 | 8/8 | 6 | 259±1 | 199±5 | 249±1 | 301±30 | 104±6 | 118±8 | 157±18 |
| KR-5 | 6 | 4.1 | 1795±87 | 1.21±0.20 | 1.28±0.21 | 8/8 | 9 | 247±6 | 187±6 | 251±1 | 273±30 | 104±6 | 117±7 | 152±18 |
| KR-9 | 6 | 4.1 | 1636±93 | 1.54±0.84 | 1.64±0.89 | 8/8 | 6 | 261±7 | 193±5 | 264±3 | 308±29 | 118±7 | 137±15 | 188±21 |
| KR-10 | 5 | 5.2 | 1778±94 | 1.63±0.28 | 1.70±0.34 | 8/8 | 10 | 274±4 | 382±15 | 283±1 | 1291±290 | 215±14 | 254±15 | >319 |
| KR-11 | 5 | 5.8 | 1896±101 | 1.63±0.28* | 1.70±0.34* | 8/8 | 9 | 330±6 | 259±9 | 345±1 | 436±35 | 137±9 | | 218±21 |
| KR-12 | 4 | 7.2 | 2056±108 | 1.63±0.28* | 1.70±0.34* | 8/8 | 15 | 343±6 | 292±16 | 357±1 | 509±169 | 142±11 | | 237±80 |
| KR-13 | 3 | 8.2 | 1371±80 | 1.66±0.28** | 1.76±0.30** | 8/8 | 12 | 344±6 | 258±11 | 357±1 | 444±78 | 188±14 | | 323±60 |
| KR-14 | 3 | 9.5 | 1630±100 | 1.66±0.28** | 1.76±0.30** | 7/8 | 15 | 310±1 | 671±29 | 295±1 | >590 | 412±31 | | >345 |
| KR-3 | 3 | 10.7 | 1528±78 | 1.63±0.28 | 1.70±0.34 | 8/8 | 24 | 295±3 | 688±62 | 317±1 | >634 | 450±47 | | >415 |
| KR-15 | 2 | 11.7 | 1085±77 | 1.20±0.18*** | 1.26±0.19*** | 7/8 | 13 | 325±1 | 602±19 | 307±1 | >618 | 555±43 | | >566 |
| KR-2 | 2 | 12.5 | 1086±77 | 1.20±0.18 | 1.26±0.19 | 8/8 | 11 | 331±4 | 681±33 | 352±1 | >704 | 626±53 | | >658 |
| KR-1 | 1 | 15.3 | 1057±77 | 1.17±0.36 | 1.32±0.38 | 8/8 | 16 | 348±4 | 683±45 | 378±2 | >756 | 647±63 | | >715 |

* g-value and $\rho$ of KR-10; ** g-value and $\rho$ of KR-3; *** g-value and $\rho$ of KR-2

[a] Fading correction after Huntley and Lamothe (2001).

[b] Fading correction after Kars et al. (2008).







**Figure 1: Location map of Kerem Shalom denoted as star (DEM from Hall, 1997) and stratigraphic section from Zilberman et al. (2007). The Negev dune field is marked in yellow. Other samples used in this study are from Shefayim (1), Ruhama (2) and the Negev Dune Field (3). Inset – Location of the coastal plain in the eastern Mediterranean.**





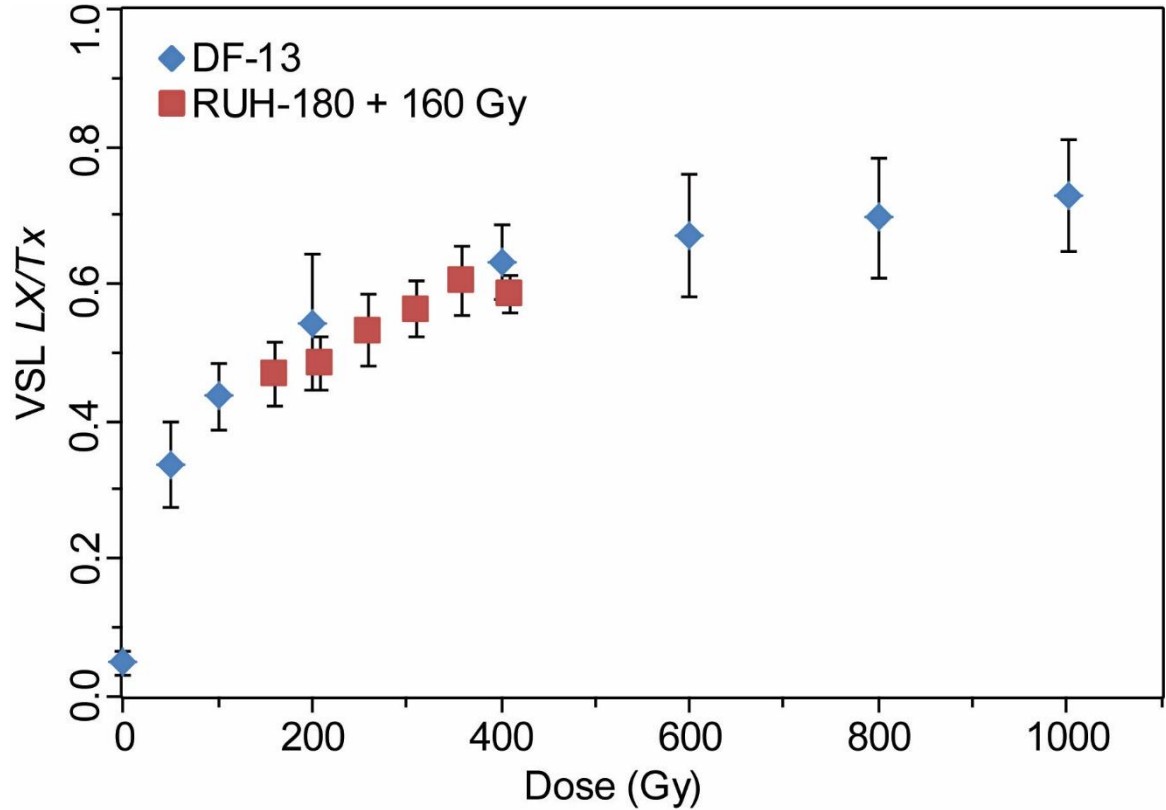

**Figure 2: VSL multiple aliquot additive dose (MAAD) DRCs of a modern sand sample (DF-13, blue diamonds) and of sample RUH-180 with addition of 160 Gy (red squares). Note that the two DRCs overlap.**







Figure 3: Representative natural luminescence signals of OSL (a), TT-OSL (b), VSL (c), and pIRIR$_{250}$ (d) of sample KR-13. The insets show the dose response curves fitted with a single exponential function. Two or three points overlap at the lowest dose point (recycling points). No dose response curve was constructed for the VSL signal. OSL signal and DRC are modified from Zilberman et al. (2007) based on measurements of 5-6 mm aliquot. Note that the pIRIR$_{250}$ $De$ is significantly lower than the TT-OSL $De$.



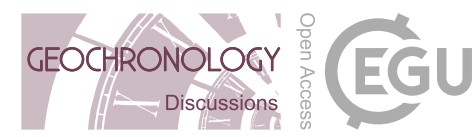

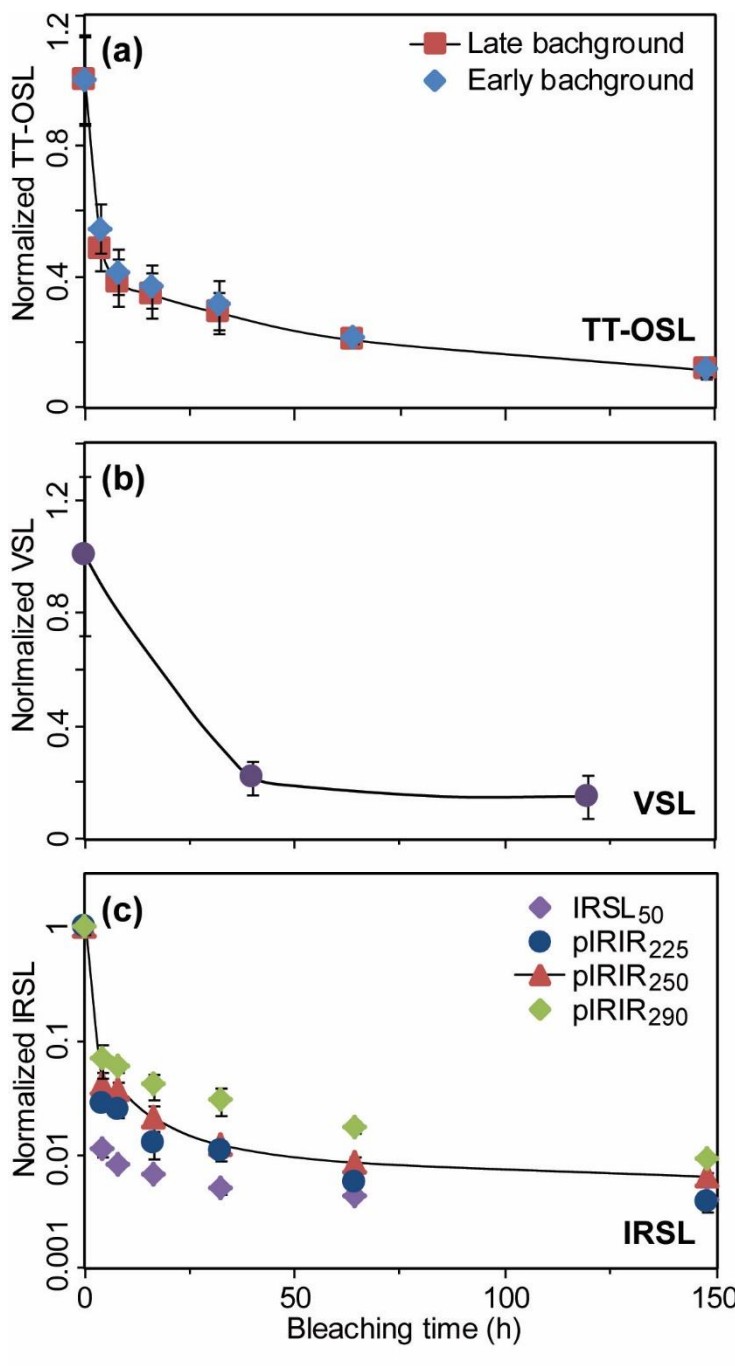

Figure 4: Bleaching experiments results for TT-OSL (a), VSL (b), and pIRIR (c) signals. Each data point is an average of 3 aliquots (4 for VSL). The TT-OSL signal was defined as first 1 s minus the following 4 s for early subtraction and first 1 s minus the last 5 s for late subtraction.



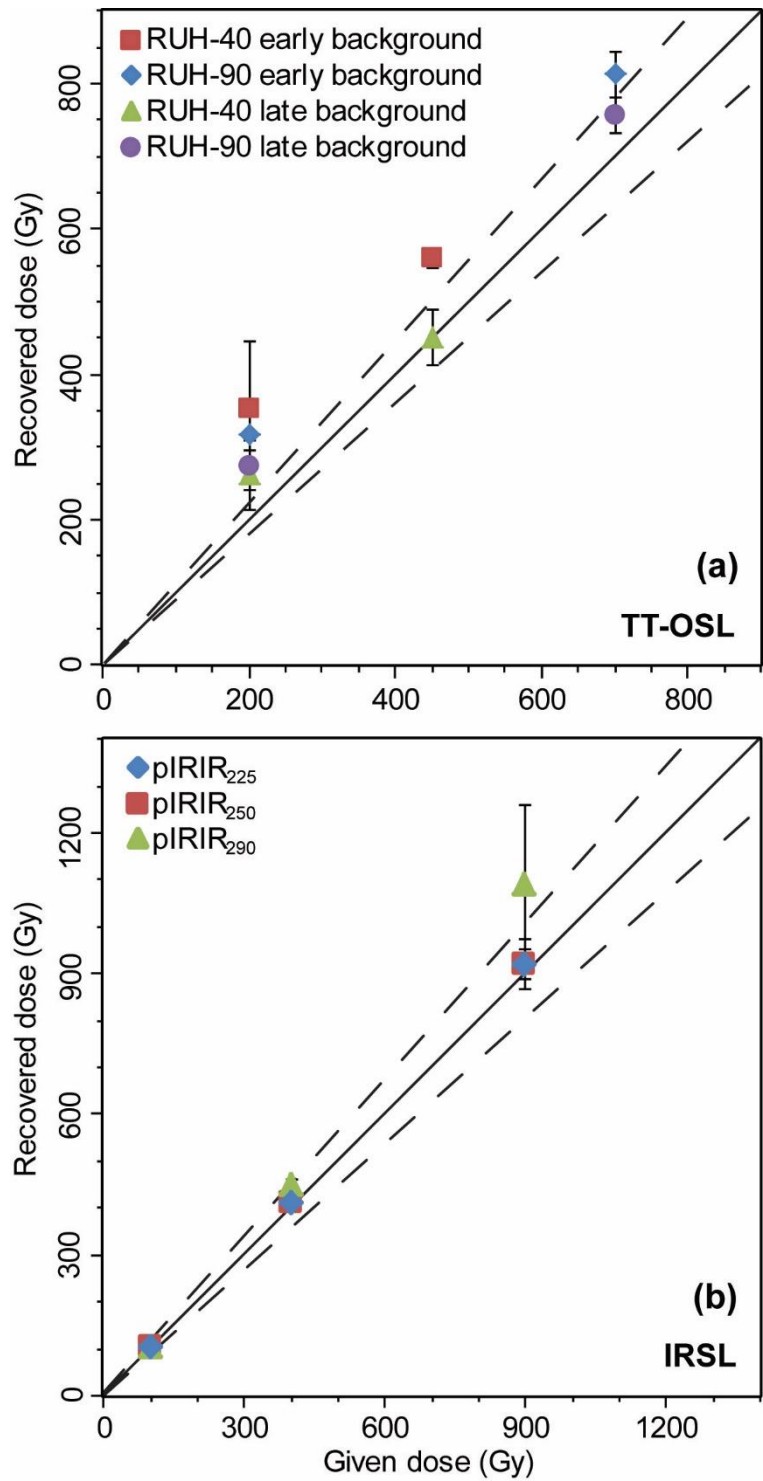

**Figure 5: Dose recovery experiments results for the TT-OSL (a) and pIRIR (b) signals. Each data point is an average of 3 or 4**
**aliquots. The solid lines are 1:1 ratio ±10% (dashed lines).**





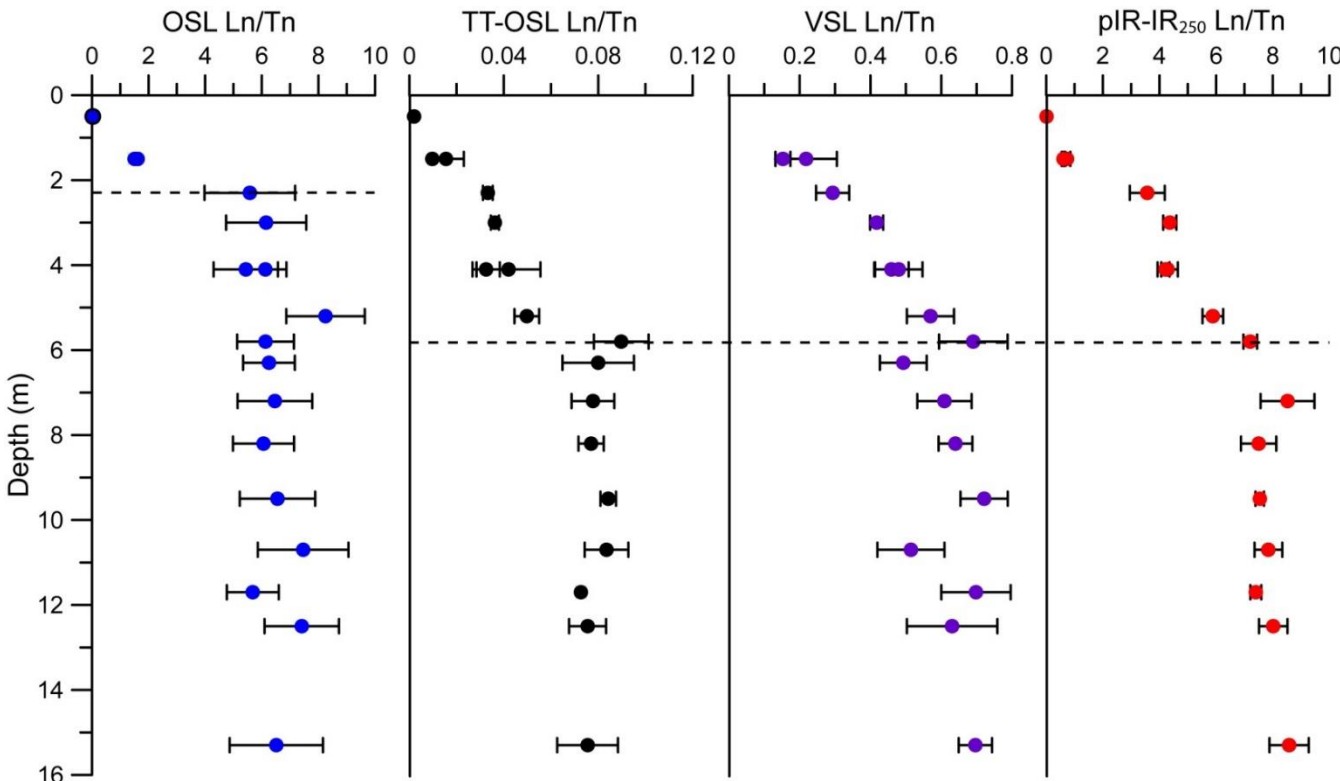

**Figure 6: Natural saturation profiles of OSL, TT-OSL, VSL, and pIRIR$_{250}$ signals. The natural luminescence signals of samples are plotted against their depth. Each data point is an average with standard deviation of 4 aliquots. OSL and TT-OSL data is modifies after Faershtein et al. (2019). The dashed lines are saturation depths of the signals.**





**Figure 7: Semi-natural DRCs of OSL, TT-OSL, and pIRIR$_{250}$.** The natural (normalized) signals of samples are plotted against their laboratory measured equivalent doses. The *Ln/Tn* values are average of four aliquots with standard deviation. OSL and TT-OSL data is modifies after Faershtein et al. (2019). Sample below the saturation depth (2 m for OSL and 6 m for TT-OSL and pIRIR$_{250}$) are in blue.




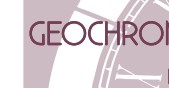 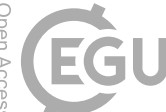


**Figure 8: TT-OSL ages and pIRIR$_{250}$ uncorrected and corrected (Huntley and Lamothe, 2001; Kars et al., 2008) ages. Some of the pIRIR$_{250}$ corrected ages after Kars et al. (2008) are minimum ages. On the right quartz environmental dose rates are presented. Note that the TT-OSL ages below 6 m mirror the dose rate pattern.**







**Figure 9: Results of the fading correction after Kars et al. (2008) for sample KR-1. Measured, unfaded, and fading corrected (simulated natural) DRCs are presented. For this sample the *Ln/Tn* is above the saturation level of the natural simulated DRC. Inset – fading rates measurement results (following Auclair et al., 2003) for this sample: *g*-value= 1.17±0.36 (% per decade) and *ρ*=1.32±0.38 (\*10⁻⁶).**





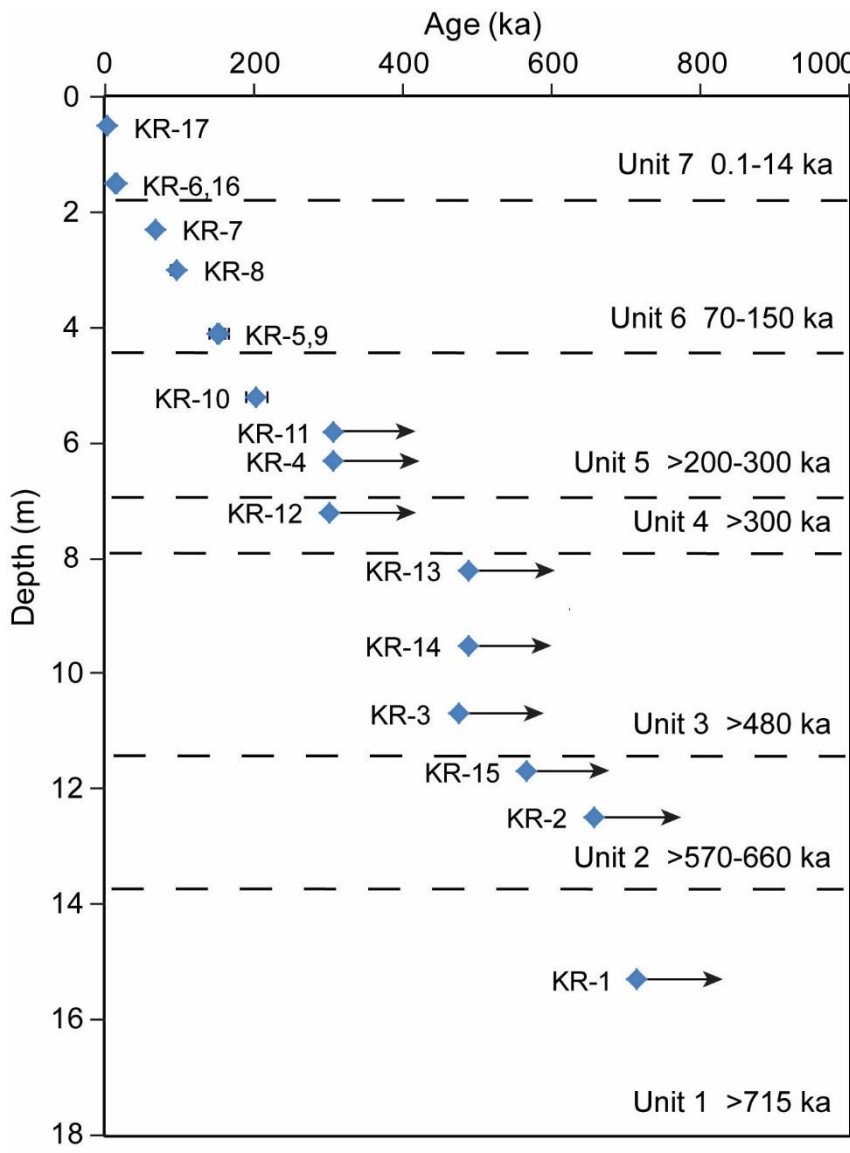

**Figure 10: KR combined luminescence ages. The ages of all samples below 6 m are minimum ages.**





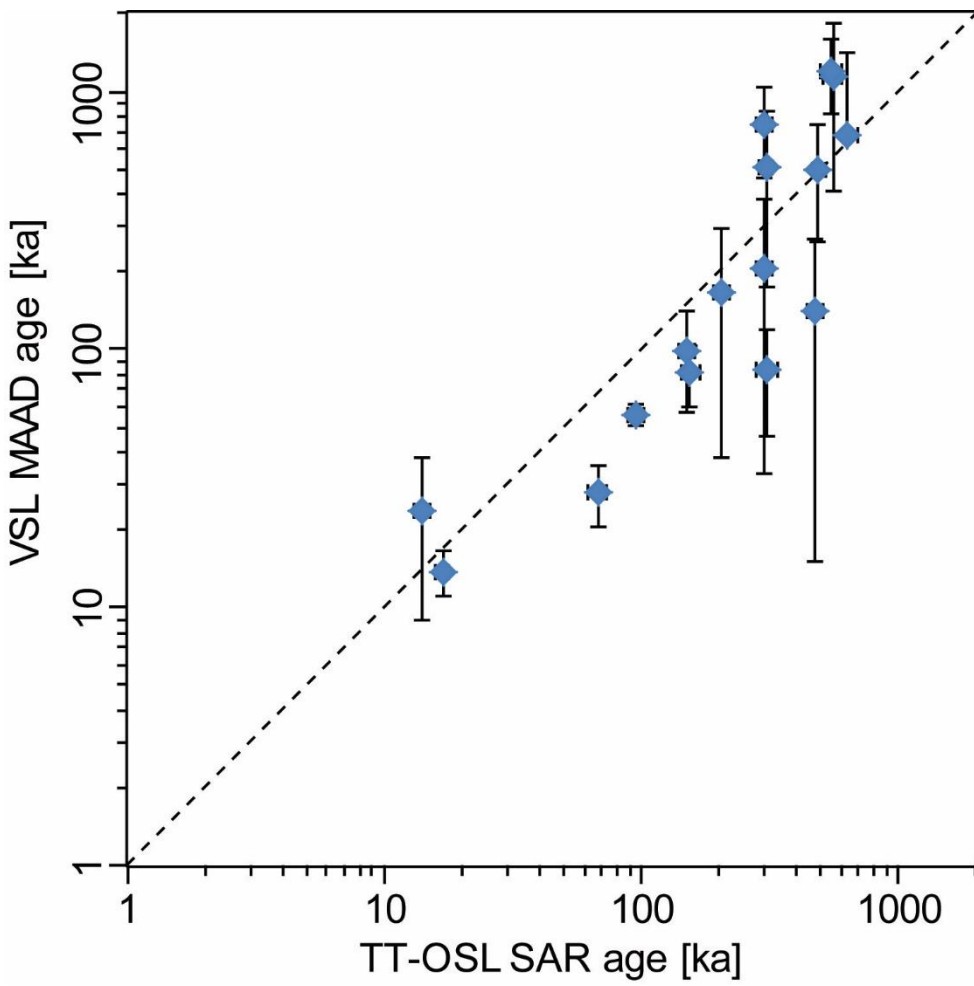

**Figure 11: VSL ages, obtained by projecting the *Ln/Tn* values of the samples on the MAAD DRC from a modern sample (DF-13), plotted against TT-OSL SAR ages. The dashed line is the 1:1 ratio.**
