# Peer review of "Alkali-feldspar separation"

_Geochronology, 2020_

## Referee Comment (RC1) · Anonymous Referee #1 · 7 Feb 2020

General comments

The authors have presented a comparison study using thermally-transferred optically stimulated luminescence (TT-OSL), violet stimulated luminescence (VSL) and post-infrared infrared stimulated luminescence (pIRIR) ages to extend the dating for the Kerem Shalom sequence from south-western Israel. This manuscript builds on their prior investigation into saturation of the natural OSL and TT-OSL signals at the site (published in Faershtein et al. 2019. Quat Geochron, 49, 146-152). The TT-OSL, VSL and pIRIR250 signals all saturate at the same depth (∼6 m), which is more likely due to the depositional environment rather than an agreement between the different

luminescence signals. This allows the authors to measure ages beyond the range of the quartz blue OSL signal (which saturates at ∼2 m depth), although all ages below ∼6m are considered minimum ages. The new minimum age for the base of the Kerem Shalom section is ∼715 ka, implying deposition via aeolian activity since the early Pleistocene.

The study is effectively motivated, and the experimental design, results, and conclusions are generally sound. This manuscript is a well-written addition to the comparative testing and application of new luminescence chronometers being developed in order to extend the age range of luminescence dating. It is suited for publication in Geochronology if the following issues are addressed.

Specific comments

In the first paragraph of the introduction, several dating methods are presented i.e. magnetostratigraphy, cosmogenics, U-Th, U-Pb and Ar-Ar dating but the only method with any supporting references is cosmogenic dating. Please add references for the other 4 methods.

In section 2: methods you mention etching the non-magnetic fraction in HF acid for 40 min. Then the alkali-feldspar is extracted from the non-magnetic fraction by heavy liquid density separation and etched with HF for 10 min. Does this mean that your feldspar samples have been etched twice? Or is the first etch for 40 min for the quartz samples and the second etch of 10 min for the feldspar samples? Either way, please clarify this.

Since you have included the wavelength of the violet laser diode, please also include the values for the blue LEDs and IR diodes. If you know the power delivered to the sample position (in mW cm-2) for the individual Riso readers that should be reported as well.

Do you think that you have grown your VSL-MAAD DRC to high enough doses to

accurately calculate the D0 value? If we are to expect comparable saturation doses for the VSL signal irrespective of location (lines 250-252), your reported $D_{0,2}$ value of 369 Gy is comparatively low. The MAAD DRC presented by Ankjaergaard 2019 continued growing beyond 8000 Gy, although it diverged from the natural DRC at $\sim$2000 Gy. When their natural DRC was fit with a double saturating exponential (DSE) function it had $D_{0,1}$ $\sim$75 Gy and $D_{0,2}$ $\sim$1300 Gy. So is it possible that the reason your data fit equally well with a SEPL or DSE fit is because you haven't extended the DRC to higher doses... Your DRC data in Fig 2 for the modern sand sample looks as though it is growing beyond 1000 Gy, although it is difficult to judge without the fitted DRC (and also because of the large uncertainties on the VSL data but there is nothing you can do about the low VSL signal intensity).

In section 3.5 you describe the TT-OSL ages as having one reversal (at 8m) and the uncorrected pIRIR ages as having two reversals (at 5 and 11 m). But looking at Tables 3 and 5 and Fig 8, there are two reversals in the TT-OSL ages (at 8m and 15m, although technically the ages at 12m and 15m are within uncertainty of each other) and only one reversal in the pIRIR ages (at 5m). Please clarify.

The final paragraph of section 3.5 becomes ambiguous because you are speaking about TT-OSL, uncorrected pIRIR, and two different versions of corrected pIRIR ages. Suggest changing line 319 slightly to remove any chance of ambiguity... "As the TT-OSL and pIRIR methods are limited by different factors (thermal and athermal signal loss respectively), there is no reason ..."

In section 3.6 please refer back to Fig. 2 (in line 328)... it's been a long time since you discussed the VSL DRC data in section 2.

In section 3.7 (lines 357-366) you discuss the effect of bioturbation as a reason for the relatively young age of sample KR-10 at the top of unit 5, which is a well-reasoned explanation. However, you then state that this phenomenon is visible also in sample KR-15 in unit 2... I think here you are over-interpreting KR-15's apparent age differ-

Interactive
comment

ence with the underlying KR-2 and suggest that you remove the last sentence in this paragraph. The relationship between ages does look similar on Fig 10 if you consider the plotted age/mid-point. But the TT-OSL and pIRIR signals are clearly saturated as shown by the Ln/Tn plot (Fig 6) and the Ln/Tn values for each signal are almost identical for these two samples. The TT-OSL De values overlap within uncertainty, so their age difference stems from variation in the quartz dose rates. In contrast, the feldspar dose rates are almost identical so the age difference is due to the different pIRIR De values. But when you compare the TT-OSL, uncorrected and corrected pIRIR ages for these two samples (Fig 8) they all overlap within uncertainty.

Following on from the previous comment, I'm not convinced about how the final chronology is presented in Fig. 10. It is not a simple task to combine multiple luminescence chronometers and your choice to use the oldest TT-OSL or pIRIR age is a good approach. But then for sample KR-14 you essentially disregard this approach and cite the principle of super position as a reason to "fix" that age between the two bracketing ages. The logic behind this argument is not necessarily incorrect but I think it glosses over the complexity of dealing with ages after signal saturation. How can you be sure that KR-13 isn't the age inversion? It has the largest variation between the different methods and although there is an hiatus between unit 3 and 4, there is another hiatus between unit 4 and 5 that is not visible in terms of the ages. If you use the uncorrected pIRIR age for KR-14 it still overlaps with the bracketing ages (KR-13 and KR-3) and it also puts into context the offset between samples KR-15 and KR-2. Ultimately, this makes the point that even when multiple luminescence signals are in saturation, the ages do not necessarily agree and can still be messy. . . which makes your Ln/Tn plots presented in Fig 6 even more impressive because they show really clearly what the ages cannot.

Fig. 2: please plot the DRC for the modern sample, preferably the single saturating exponential plus linear fit, as that is the fit ultimately used in the age comparison in Fig 11.

Fig 4. Why are there so few points reported on the bleaching experiment for the VSL signal (n=3)? The bleaching signal is much better defined for the TT-OSL and pIRIR experiments (n=7). Additional measurements should be made to better define the VSL signal bleaching curve.

Fig 10. Suggest adding a short explanation to the text about which ages are used in this figure. For example, "ages above 6m are based on uncorrected pIRIR250, while ages below 6m are the oldest TT-OSL or corrected pIRIR250"

Fig 11. Please use different symbols/colours for the samples above and below the 6m saturation cut-off to make the comparison easier to see.

Technical corrections

Line 24: change semi-colon (after reversals) to full stop

Line 32: change to "...(quartz or alkali-feldspar)..."

Line 55: remove "the" before luminescence methods

Line 59: the reference Haler et al. 2017 does not appear in the reference list, I assume this is a typo and is meant to be Harel et al. 2017

Line 69: change unites to units

Line 82: change experiment to experiments

Line 86: suggest changing this to "...from the KR section by drilling 30cm deep holes horizontally into the sediment".

Line 156: change Recycling to recycling

Line 164: change reduces to reduced

Line 190: change to "Previous studies reported lower residual signals..."

Line 198: remove hyphen from pIR-IR as it is not consistent with the rest of the

manuscript

Line 199: changed droped to dropped

Line 233: change sample's to sample

Line 238: remove "Similarly" and begin sentence "The natural TT-OSL . . ."

Line 280: remove "the" so sentence reads "The natural growth signal is limited by anomalous fading"

Line 285: Change Huntly to Huntley

Line 309: change aged to ages

Line 334: remove farther

Line 367: The depositional ages reported for the units do not match with those reported in Fig 10. Please clarify.

Line 395: remove repeated "to" text should read ". . .later penetration of the silt into the sandy soil"

Line 470: change oncentrations to concentrations

Table 4: change samples to sample, text should read "For sample details see Table S2"

Fig. 4a legend: change bachground to background

Fig 5 caption: change experiments to experimental

Fig 6. caption: change modifies to modified

Fig 7. caption: change modifies to modified

Table S3 caption: change exponentials to exponential

Table S3: please order samples by depth (rather than sample ID) in keeping with the

rest of the manuscript

---

## Referee Comment (RC2) · Anonymous Referee #2 · 26 Feb 2020

Dear Authors and Editor,

This manuscript presents an interesting comparison of three long-range luminescence chronometers: TT-OSL, VSL, and pIRIRSL. While the authors do determine age saturation for the oldest sediments, they also extend the existing depositional chronology at this site. Overall the manuscript reads well, is scientifically robust and presents novel results. I have provided some comments below which I hope will prove useful to the authors.

538-40: It might be clearer to list the references with the technique list: "(TT-OSL; Wang et al., 2006a)..."

42-48: Other primary limitations include a low signal-to-noise ratio (TT-OSL) and the long period of time required to bleach all three signals relative to conventional BLOSL.

78: Capitalize "K"

83: Unclear here whether TT-OSL and pIRIR signals are dated with SAR or MAAD protocol. Please rephrase for clarity.

112: Please list what is meant by 'sensitized aliquots.' Are these simply discs that have been through the SAR cycle, or does this mean something else?

112-116: It would be nice to show the fading data (and fitted functions for g and rho). Could you include these in the supplement please? Also, to be clear, is the fitted value rho or rho'? From the caption of Fig. 9 it seems like rho'.

125-127: Please justify why this approach is preferred. If another sample were more variable in DRC or Ln/Tn values, one might expect an approach like this to produce bias.

140: I am confused by this sentence. Maybe change from "is comparable" to "should be comparable" as it seems that you are referencing another dataset. I am unclear on the meaning of this statement: "...it is expected that DRCs constructed for different samples would be comparable as well."

Comparable to each other? Comparable to the natural DRC? Both?

Also, "the MAAD DRC is comparable to the natural DRC" is a bit ambiguous. Does this refer specifically to your DF-13 data? And are you comparing your data against data from Ankjaergaard? Are you interpreting DF-13 data with the help of conclusions from Ankjaergaard? Or are you simply restating a conclusion of Ankjaergaard? Please clarify.

145: My understanding is that the 160 Gy added to RUH-180 is not an actual dose given in the instrument, but rather a number added to the x-coordinate of the data.

While I think this is a clever thing to do (and really like your suggestion of treating these data similar to RF), neither Fig. 2 nor the text make this clear. For the text, please clarify that the data are shifted but the actual given doses range from 0 to 200? Gy. Likewise, Fig. 2 should be reworked to avoid the false impression that the samples were given doses of 160 to 400? Gy. Perhaps the use of an arrow, or a secondary inset x-scale for the red boxes.

165-166: "saturates at 700-800 Gy" Is this D0? 2D0? "saturates" in this context is an ambiguous concept.

234ff: While I basically agree with your assessment that saturation occurs around 2 m and 6 m for OSL and the others, it might help to be slightly more quantitative, if possible. For example, why not saturation at 4 m for VSL? That datapoint has 1-sigma overlap with the lowest sample in that profile.

261: A third option would be significant erosion which strips off material down and exposes the old, saturated units. This seems incompatible with your 'clay from the surface' hypothesis though.

283: "fading rates increase over geological time" I'm confused by this statement. The functional form of both the Huntley and Lamothe (2001) and the Kars et al. (2008) would yield the opposite response following a lab dose–a decreasing rate through time–either as a simple logarithmic decay or as a sigmoid (in log-x space). If instead you mean that fading rate should increase with geologic dose, i.e., that only unstable sites remain open, then that makes sense. But how this relates to your argument is not clear to me.

296-298: Here and in Fig. 8, I think the argument that 'age mirrors dose rate' is a little misleading. Earlier in the manuscript you seem to indicate that samples below 6 m are close to saturation for TT-OSL and that these TT-OSL ages should be treated as minimum ages. If this is the case, then a) this should be clear in Fig. 8 (currently there is no indication that TT-OSL samples below 6 m are minimum ages), and b) the

comparision you really make is between dose rate and 2D0/Ddot or similar (e.g., time until near saturation). This relationship is informative for characterizing samples but not for providing a depositional timeline, as would currently be interpreted from Fig. 8. Fi.g 10 does a better job at representing this.

319: "signal loss"

327-328: How similar were the growth curves of the KR samples? Was this examined in order to justify using a common MAAD curve for all KR natural signals?

359: "one can expect the A and upper B horizons to be kept relatively bleached all the time." This may be the case, but the portion of grains that are fully bleached due to bioturbation is likely to depend upon the local plants and animals.

Tables 1, 3, 5: Unconventional to give dose rates as microGy/a. Please consider using milliGy/a instead (better yet, Gy/ka, given that ages are reported in ka and doses in Gy).

Fig. 3: "OSL signal and DRC are modified from Zilberman et al. (2007)" Please describe this modification, here or in the main text.

Fig. 9: Are the Ln/Tn error bars shown? Please include these if not.

---

## Short Comment (SC1) · 26 Feb 2020

The effort to elaborate a luminescence dating technique that allows determining accurate mid-Pleistocene ages is highly appreciated. The topic is indeed of prime interest to many of us and I have looked at this paper with great curiosity. I got a little lost between ages, depth and sediment units, but finally found out that the extended range stops at around 150 ka. My question is why standard quartz OSL is unable to determine an age of this range? The sediment description suggests that a pedogenetically altered dune sand has been dated. This dune sand should be composed of low activity quartz (alongside some plant and snail remains). The dose rate should therefore be <1

[Figure]

Gy/ka and this is indeed so in the lowermost part of the section where K activity is low as expected. In semi-arid environments pedogenesis leads to uranium leaching down profile, typically alongside carbonate leaching (Langmuir 1978, Geochim. Cosmochim. Acta 42, 547). The latter is clearly evident from the Zilberman et al report (see Figs 5,6,7 therein), but it is less clear from the U/Th ratios listed in Table 1. Thus, uranium mobilisation was indeed negligible or, more likely, the disturbance terminated some U-234 half-lives ago, or, equally likely, some layers are enriched in Ra-226. I suspect that the dose rate was not constant during burial.

---

## Author Comment (AC1) · 17 Mar 2020

We thank the anonymous referee #1 for the positive and constructive review and we address specific comments below.

In the first paragraph of the introduction, several dating methods are presented i.e. magnetostratigraphy, cosmogenics, U-Th, U-Pb and Ar-Ar dating but the only method with any supporting references is cosmogenic dating. Please add references for the other 4 methods.

The references were added.

[Figure]

In section 2: methods you mention etching the non-magnetic fraction in HF acid for 40 min. Then the alkali-feldspar is extracted from the non-magnetic fraction by heavy liquid density separation and etched with HF for 10 min. Does this mean that your feldspar samples have been etched twice? Or is the first etch for 40 min for the quartz samples and the second etch of 10 min for the feldspar samples? Either way, please clarify this.

The feldspar samples were etched only once, for 10 minutes with 10% HF solution. This was clarified in the manuscript.

Since you have included the wavelength of the violet laser diode, please also include the values for the blue LEDs and IR diodes. If you know the power delivered to the sample position (in mW cm-2) for the individual Riso readers that should be reported as well.

The wavelengths and the power delivered to the sample were added for three light sources.

Do you think that you have grown your VSL-MAAD DRC to high enough doses to accurately calculate the D0 value? If we are to expect comparable saturation doses for the VSL signal irrespective of location (lines 250-252), your reported D0,2 value of 369 Gy is comparatively low. The MAAD DRC presented by Ankjaergaard 2019 continued growing beyond 8000 Gy, although it diverged from the natural DRC at _2000 Gy. When their natural DRC was fit with a double saturating exponential (DSE) function it had D0,1 _75 Gy and D0,2 _1300 Gy. So is it possible that the reason your data fit equally well with a SEPL or DSE fit is because you haven't extended the DRC to higher doses. . . Your DRC data in Fig 2 for the modern sand sample looks as though it is growing beyond 1000 Gy, although it is difficult to judge without the fitted DRC (and also because of the large uncertainties on the VSL data but there is nothing you can do about the low VSL signal intensity).

We constructed our VSL MAAD DRC to 1000 Gy. It is possible that the DRC was
not constructed up to high enough doses and it would reach saturation level at higher doses. Unfortunately, additional measurements cannot be done due to lack of material. It was clarified in the manuscript that the difference between our and Ankjaergaard's D0 values may be attributed to the unsaturated VSL MAAD DRC.

In section 3.5 you describe the TT-OSL ages as having one reversal (at 8m) and the uncorrected pIRIR ages as having two reversals (at 5 and 11 m). But looking at Tables 3 and 5 and Fig 8, there are two reversals in the TT-OSL ages (at 8m and 15m, although technically the ages at 12m and 15m are within uncertainty of each other) and only one reversal in the pIRIR ages (at 5m). Please clarify.

This could have been a misunderstanding. There is one significant reversal at the TT-OSL ages (at 8 m). At 15 m the ages are agree within error; therefore, we don't consider this a reversal. Regarding the uncorrected pIRIR250 ages, there is indeed only one reversal. This was corrected in the manuscript.

The final paragraph of section 3.5 becomes ambiguous because you are speaking about TT-OSL, uncorrected pIRIR, and two different versions of corrected pIRIR ages. Suggest changing line 319 slightly to remove any chance of ambiguity. . . "As the TTOSL and pIRIR methods are limited by different factors (thermal and athermal signal loss respectively), there is no reason . . ."

Done.

In section 3.6 please refer back to Fig. 2 (in line 328). . . it's been a long time since you discussed the VSL DRC data in section 2.

Done.

In section 3.7 (lines 357-366) you discuss the effect of bioturbation as a reason for the relatively young age of sample KR-10 at the top of unit 5, which is a well-reasoned explanation. However, you then state that this phenomenon is visible also in sample KR-15 in unit 2. . . I think here you are over-interpreting KR-15's apparent age difference with the underlying KR-2 and suggest that you remove the last sentence in this paragraph. The relationship between ages does look similar on Fig 10 if you consider the plotted age/mid-point. But the TT-OSL and pIRIR signals are clearly saturated as shown by the Ln/Tn plot (Fig 6) and the Ln/Tn values for each signal are almost identical for these two samples. The TT-OSL De values overlap within uncertainty, so their age difference stems from variation in the quartz dose rates. In contrast, the feldspar dose rates are almost identical so the age difference is due to the different pIRIR De values. But when you compare the TT-OSL, uncorrected and corrected pIRIR ages for these two samples (Fig 8) they all overlap within uncertainty.

We agree with the referee. The sentence was deleted.

Following on from the previous comment, I'm not convinced about how the final chronology is presented in Fig. 10. It is not a simple task to combine multiple luminescence chronometers and your choice to use the oldest TT-OSL or pIRIR age is a good approach. But then for sample KR-14 you essentially disregard this approach and cite the principle of super position as a reason to "fix" that age between the two bracketing ages. The logic behind this argument is not necessarily incorrect but I think it glosses over the complexity of dealing with ages after signal saturation. How can you be sure that KR-13 isn't the age inversion? It has the largest variation between the different methods and although there is an hiatus between unit 3 and 4, there is another hiatus between unit 4 and 5 that is not visible in terms of the ages. If you use the uncorrected pIRIR age for KR-14 it still overlaps with the bracketing ages (KR-13 and KR-3) and it also puts into context the offset between samples KR-15 and KR-2. Ultimately, this makes the point that even when multiple luminescence signals are in saturation, the ages do not necessarily agree and can still be messy. . . which makes your Ln/Tn plots presented in Fig 6 even more impressive because they show really clearly what the ages cannot.

We accept the point presented by the referee. The final age of sample KR-14 was defined as minimum Kars corrected pIRIR250 age without the superposition correction.

Figure 10 was corrected accordingly.

Fig. 2: please plot the DRC for the modern sample, preferably the single saturating exponential plus linear fit, as that is the fit ultimately used in the age comparison in Fig 11.

Done.

Fig 4. Why are there so few points reported on the bleaching experiment for the VSL signal (n=3)? The bleaching signal is much better defined for the TT-OSL and pIRIR experiments (n=7). Additional measurements should be made to better define the VSL signal bleaching curve.

Due to lack of material only two bleaching times (along with 0 h bleaching point) were measured as part of the VSL bleaching experiment. Unfortunately, no additional measurements can be made. Although the definition of signal bleaching is not optimal, the general trend is clear.

Fig 10. Suggest adding a short explanation to the text about which ages are used in this figure. For example, "ages above 6m are based on uncorrected pIRIR250, while ages below 6m are the oldest TT-OSL or corrected pIRIR250".

Done.

Fig 11. Please use different symbols/colours for the samples above and below the 6m saturation cut-off to make the comparison easier to see.

Done.

Technical corrections Line 24: change semi-colon (after reversals) to full stop.

Done.

Line 32: change to ". . .(quartz or alkali-feldspar). . .".

Done.

Line 55: remove "the" before luminescence methods.

Done.

Line 59: the reference Haler et al. 2017 does not appear in the reference list, I assume this is a typo and is meant to be Harel et al. 2017.

Changed to "Harel et al., 2017".

Line 69: change unites to units.

Done.

Line 82: change experiment to experiments.

Done.

Line 86: suggest changing this to ". . .from the KR section by drilling 30cm deep holes horizontally into the sediment".

Done.

Line 156: change Recycling to recycling.

Done.

Line 164: change reduces to reduced.

Done.

Line 190: change to "Previous studies reported lower residual signals. . .".

Done.

Line 198: remove hyphen from pIR-IR as it is not consistent with the rest of the manuscript.

Done.

Line 199: changed droped to dropped.

Done.

Line 233: change sample's to sample.

Done.

Line 238: remove "Similarly" and begin sentence "The natural TT-OSL . . .".

Done.

Line 280: remove "the" so sentence reads "The natural growth signal is limited by anomalous fading".

Done.

Line 285: Change Huntly to Huntley.

Done.

Line 309: change aged to ages.

Done.

Line 334: remove farther.

Done.

Line 367: The depositional ages reported for the units do not match with those reported in Fig 10. Please clarify.

In the text the ages were rounded for simplicity. We recognize that the difference may be confusing. Therefore, the ages were corrected for exact minimum ages to fit Fig. 10.

Line 395: remove repeated "to" text should read ". . .later penetration of the silt into the sandy soil".

Done.

Line 470: change oncentrations to concentrations.

Done.

Table 4: change samples to sample, text should read "For sample details see Table S2".

Done.

Fig. 4a legend: change bachground to background.

Done.

Fig 5 caption: change experiments to experimental.

Done.

Fig 6. caption: change modifies to modified.

Done.

Fig 7. caption: change modifies to modified.

Done.

Table S3 caption: change exponentials to exponential.

Done.

Table S3: please order samples by depth (rather than sample ID) in keeping with the rest of the manuscript.

Done.

---

## Author Comment (AC2) · 17 Mar 2020

We thank the anonymous referee #2 for the positive and constructive review and address specific comments below.

38-40: It might be clearer to list the references with the technique list: "(TT-OSL; Wang et al., 2006a)..."

Done.

42-48: Other primary limitations include a low signal-to-noise ratio (TT-OSL) and the long period of time required to bleach all three signals relative to conventional BLOSL.

[Figure]

In the manuscript we present some examples for limitations of the different extended range methods. The referee suggests adding additional limitations. In our respectful opinion, the main idea is clear without adding more disadvantages of these methods. In addition, the signal to noise ratio of the TT-OSL varies from sample to sample and we don't see is as a limiting factor.

78: Capitalize "K"

Done.

83: Unclear here whether TT-OSL and pIRIR signals are dated with SAR or MAAD protocol. Please rephrase for clarity.

The TT-OSL and pIRIR250 ages are obtained with SAR protocol. It was clarified in the manuscript.

112: Please list what is meant by 'sensitized aliquots.' Are these simply discs that have been through the SAR cycle, or does this mean something else?

These are discs that went through several SAR cycles. It was clarified in the manuscript.

112-116: It would be nice to show the fading data (and fitted functions for g and rho). Could you include these in the supplement please? Also, to be clear, is the fitted value rho or rho'? From the caption of Fig. 9 it seems like rho'.

The average g-values and 's are presented in Table 5. In addition, fading data of all measured aliquots was added to the supplementary.

125-127: Please justify why this approach is preferred. If another sample were more variable in DRC or Ln/Tn values, one might expect an approach like this to produce bias.

We applied the calc_Huntley2006 R function using average Ln/Tn and the combined DRCs for all samples. As explained in the manuscript, this approach resulted in almost identical results as when applying the function on all measured aliquots (0-4 % difference). In our opinion, the variability on the DRC and Ln/Tn is embodied in the combined DRC and average Ln/Tn. Therefore, this approach is appropriate and time saving. The comparison made for sample KR-1 was added to the supplementary.

140: I am confused by this sentence. Maybe change from "is comparable" to "should be comparable" as it seems that you are referencing another dataset. I am unclear on the meaning of this statement: "...it is expected that DRCs constructed for different samples would be comparable as well." Comparable to each other? Comparable to the natural DRC? Both? Also, "the MAAD DRC is comparable to the natural DRC" is a bit ambiguous. Does this refer specifically to your DF-13 data? And are you comparing your data against data from Ankjaergaard? Are you interpreting DF-13 data with the help of conclusions from Ankjaergaard? Or are you simply restating a conclusion of Ankjaergaard? Please clarify.

Ankjærgaard et al. (2016) showed that a MAAD constructed DRC is comparable to a combined natural DRC for the Chines loess. Following these results, it is expected that MAAD DRCs of different samples of the same source would be comparable. The sentence was rephrased for clarity.

145: My understanding is that the 160 Gy added to RUH-180 is not an actual dose given in the instrument, but rather a number added to the x-coordinate of the data. While I think this is a clever thing to do (and really like your suggestion of treating these data similar to RF), neither Fig. 2 nor the text make this clear. For the text, please clarify that the data are shifted but the actual given doses range from 0 to 200? Gy. Likewise, Fig. 2 should be reworked to avoid the false impression that the samples were given doses of 160 to 400? Gy. Perhaps the use of an arrow, or a secondary inset x-scale for the red boxes.

Sample RUH-180 received doses of 0 to 200 Gy. The data points of this sample were shifted by 160 Gy on figure 2. This was clarified in the manuscript and the figure

caption.

165-166: "saturates at 700-800 Gy" Is this D0? 2D0? "saturates" in this context is an ambiguous concept.

The laboratory DRC of pIRIR250 reaches 2D0 at 700-800 Gy. This was clarified in the manuscript.

234ff: While I basically agree with your assessment that saturation occurs around 2 m and 6 m for OSL and the others, it might help to be slightly more quantitative, if possible. For example, why not saturation at 4 m for VSL? That datapoint has 1-sigma overlap with the lowest sample in that profile.

The concept of saturation profiles (plotting natural signals vs. depth) is qualitative due to variation in dose rates. Fitting the data with a certain function would by misleading. For the VSL signal the saturation level is harder to determine as this signal is rather noisy. Although the Ln/Tn level at 4 m overlaps with the Ln/Tn levels at 6 m and 11 m it dose not overlap with the lowermost sample (15 m). Discussion regarding the saturation depth of VSL was added to the manuscript.

261: A third option would be significant erosion which strips off material down and exposes the old, saturated units. This seems incompatible with your 'clay from the surface' hypothesis though.

This option was added to the manuscript.

283: "fading rates increase over geological time" I'm confused by this statement. The functional form of both the Huntley and Lamothe (2001) and the Kars et al. (2008) would yield the opposite response following a lab dose–a decreasing rate through time–either as a simple logarithmic decay or as a sigmoid (in log-x space). If instead you mean that fading rate should increase with geologic dose, i.e., that only unstable sites remain open, then that makes senese. But how this relates to your argument is not clear to me.

Fading rates can increase at high absorbed doses as was demonstrated by Huntley and Lian (2006) and Wallinga et al. (2007). This process is considered by the fading correction of Kars et al. (2008).

296-298: Here and in Fig. 8, I think the argument that 'age mirrors dose rate' is a little misleading. Earlier in the manuscript you seem to indicate that samples below 6 m are close to saturation for TT-OSL and that these TT-OSL ages should be treated as minimum ages. If this is the case, then a) this should be clear in Fig. 8 (currently there is no indication that TT-OSL samples below 6 m are minimum ages), and b) the comparision you really make is between dose rate and 2D0/Ddot or similar (e.g., time until near saturation). This relationship is informative for characterizing samples but not for providing a depositional timeline, as would currently be interpreted from Fig. 8. Fi.g 10 does a better job at representing this.

As explained in the manuscript TT-OSL De values of all samples below 6 m cluster at 400-500 Gy but resulting in apparent TT-OSL age increase with depth due to decrease in the environmental dose rates. The ages discuses in the text and presented in figure 8 are calculated ages (Table 3) and not 2D0 ages. Hence, comparison to the dose rate trend is adequate. It was added in the caption of figure 8 that all ages below 6 m should be treated as minimum.

319: "signal loss"

Done.

327-328: How similar were the growth curves of the KR samples? Was this examined in order to justify using a common MAAD curve for all KR natural signals?

Interpolation of natural signals onto a MAAM DRC of a modern sample was done based on the results of Ankjærgaard et al. (2016), who showed that a MAAM DRC of a modern sample is very close to the natural DRC of their samples. Laboratory VSL DRCs were not constructed for the KR samples as discussed in the manuscript. Nevertheless, comparison of MAAD DRCs of two samples is presented in the manuscript (Sect. 2; figure 2), strengthening the interpolation approach.

359: "one can expect the A and upper B horizons to be kept relatively bleached all the time." This may be the case, but the portion of grains that are fully bleached due to bioturbation is likely to depend upon the local plants and animals.

That could be true but cannot really be quantified. As a first-order approximation we assumed that vegetation type and cover did not change significantly over time.

Tables 1, 3, 5: Unconventional to give dose rates as microGy/a. Please consider using milliGy/a instead (better yet, Gy/ka, given that ages are reported in ka and doses in Gy).

The dose rates were changed to Gy ka-1 in all tables.

Fig. 3: "OSL signal and DRC are modified from Zilberman et al. (2007)" Please describe this modification, here or in the main text.

The signal and DRC were not modified. This was corrected in the figure caption.

Fig. 9: Are the Ln/Tn error bars shown? Please include these if not.

The error bars for the Ln/Tn are shown in the figure.

---

## Author Response (AR2)

Dear Editor (Julie Durcan),

Thank you for the positive response.

We took into consideration Dr. Mauz's note about the navigation between the different ages, units, and depths and refined our manuscript to be more clear. We are also unhappy that the OSL method was found to be limited to 100 Gy for this section. The other methods used in our study enable dating up to ages corresponding to 400 and 600 Gy (for quartz and feldspar respectively). Hopefully, in the future older sediments could be accurately dated using the luminescence methods.

Regarding changes in dose rate with time – we can only measure the present dose rates and get a snapshot in time. Pedogenic processes can be evaluated but it is difficult to translate that into specific changes in dose rates over time. The reason some samples have higher dose rate is because they are rich with silt and clay. In any case, changes in the dose rates over time would not change significantly the results of our study.

Best regards,

Galina Faershtein

[revised manuscript text omitted]
. This value is significantly lower than the $D_0$ value obtained by Ankjærgaard et al. (2019) for a combined natural DRC from Chinese loess samples ($D_{0,2}$=1334±504 Gy for a double saturating exponential with a constant vertical offset). It is possible that the MAAD DRC constructed here dose not reach saturation, resulting in lower $D_0$ value.

Based on the results of Ankjærgaard et al. (2016), which suggest thatAs the MAAD DRC is comparable to the natural DRC, (Ankjærgaard et al., 2016), it is expected that MAAD DRCs constructed for different samples (of the same source) would be comparable to each other 
[revised manuscript text omitted]
|---|---|---|---|---|---|---|---|---|---|---|---|
| KR-17 | 7 | 0.5 | 88-125 | 0.73 | 0.8 | 2.4 | 0.004 | 0.641 | 0.597 | 1.24±0.076 | 1.59±0.09 |
| KR-6 | 7 | 1.5 | 125-150 | 0.73 | 0.68 | 2.15 | 0.003 | 0.611 | 0.604 | 1.22±0.076 | 1.69±0.08 |
| KR-16 | 7 | 1.5 | 88-125 | 0.75 | 0.7 | 2.2 | 0.003 | 0.637 | 0.560 | 1.20±0.066 | 1.56±0.098 |
| KR-7 | 6 | 2.3 | 74-105 | 0.83 | 1.8 | 4.2 | 0.009 | 0.880 | 0.709 | 1.60±0.08 | 1.90±0.09 |
| KR-8 | 6 | 3 | 88-125 | 0.77 | 2.3 | 2.6 | 0.008 | 0.858 | 0.690 | 1.56±0.08 | 1.91±0.10 |
| KR-5 | 6 | 4.1 | 125-150 | 0.73 | 0.88 | 2.01 | 0.003 | 0.632 | 0.684 | 1.32±0.077 | 1.80±0.098 |
| KR-9 | 6 | 4.1 | 88-125 | 0.68 | 0.9 | 2.5 | 0.004 | 0.622 | 0.606 | 1.23±0.076 | 1.64±0.09 |
| KR-10 | 5 | 5.2 | 75-105 | 0.59 | 1.5 | 3.5 | 0.008 | 0.665 | 0.690 | 1.36±0.077 | 1.78±0.09 |
| KR-11 | 5 | 5.8 | 88-125 | 0.7 | 1.8 | 4.4 | 0.008 | 0.791 | 0.739 | 1.54±0.08 | 1.90±0.10 |
| KR-4 | 5 | 6.3 | 125-150 | 0.76 | 1.73 | 3.99 | 0.006 | 0.801 | 0.753 | 1.56±0.08 | - |
| KR-12 | 4 | 7.2 | 88-125 | 0.73 | 2.2 | 4.5 | 0.009 | 0.863 | 0.826 | 1.70±0.098 | 2.06±0.11 |
| KR-13 | 3 | 8.2 | 88-125 | 0.38 | 1.7 | 2.8 | 0.007 | 0.529 | 0.478 | 1.014±0.054 | 1.37±0.08 |
| KR-14 | 3 | 9.5 | 88-125 | 0.45 | 1.4 | 2.4 | 0.005 | 0.529 | 0.739 | 1.27±0.087 | 1.63±0.10 |
| KR-3 | 3 | 10.7 | 125-150 | 0.38 | 1.3 | 2.64 | 0.004 | 0.469 | 0.579 | 1.052±0.066 | 1.53±0.078 |
| KR-15 | 2 | 11.7 | 88-125 | 0.29 | 0.9 | 1.7 | 0.004 | 0.344 | 0.382 | 0.730±0.044 | 1.09±0.087 |
| KR-2 | 2 | 12.5 | 125-150 | 0.29 | 0.62 | 1.28 | 0.002 | 0.295 | 0.383 | 0.680±0.044 | 1.09±0.087 |
| KR-1 | 1 | 15.3 | 125-150 | 0.27 | 0.56 | 1.49 | 0.002 | 0.280 | 0.398 | 0.680±0.054 | 1.06±0.087 |

640

**Table 2: Measurement protocols and details of quartz TT-OSL and VSL and feldspar pIRIR signals used in this study.**

| | TT-OSL | VSL | pIR-IR$_{225/250/290}$ |
|---|---|---|---|
| **Discs/cups** | Aluminum discs | Aluminum discs | Stainless steel cups |
| **Aliquot size** | 5 mm | 5 mm | 1-2 mm |
| **Signal & background** | First 1 s & last 5 s | First 3 s & last 30 s | First 1 s & last 10 s |
| **Step** | | | |
| **1** | $\beta$ dose | $\beta$ dose | $\beta$ dose |
| **2** | TL at 260 °C for 10 s | TL at 300 °C for 100 s | TL at 255/280/320 °C for 60 s |
| **3** | Blue stimulation at 125 °C for 300 s | Blue stimulation at 125 °C for 100 s | IR stimulation at 50 °C for 200 s |
| **4** | TL at 260 °C for 10 s | | |
| **5** | Blue stimulation at 125 °C for 100 s (*Lx*) | VSL at 30 °C for 500 s (*Lx*) | IR stimulation at 225/250/290 °C for 200 s (*Lx*) |
| **6** | Test dose (2.2 Gy) | Test dose (490 Gy) | Test dose (30 Gy) |
| **7** | TL at 220 °C for 10 s | TL at 290 °C for 100 s | TL at 255/280/320 °C for 60 s |
| **8** | | Blue stimulation at 125 °C for 100 s | IR stimulation at 50 °C for 200 s |
| **9** | Blue stimulation at 125 °C for 100 s (*Tx*) | VSL at 30 °C for 500 s (*Tx*) | IR stimulation at 225/250/290 °C for 200 s (*Tx*) |
| **10** | Heat at 350 °C for 100 s | VSL at 380 °C for 200 s | IR stimulation at 350 °C for 300 s |

Table 3: TT-OSL dating results of the Kerem Shalom samples. No. aliquots – number of aliquots used for *De* determination out of those measured. Average *De* values and errors were calculated using the CAM after removing distinct outliers (Galbraith and Roberts., 2012). OSL *De* and ages are from Zilberman et al. (2007) for comparison.

[revised manuscript text omitted]

670 **Figure 3: Representative natural luminescence signals of OSL (a), TT-OSL (b), VSL (c), and pIRIR$_{250}$ (d) of sample KR-13 . The insets show the dose response curves fitted with a single exponential function. Two or three points overlap at the lowest dose point (recycling points). No dose response curve was constructed for the VSL signal. OSL signal and DRC are  from data of Zilberman et al. (2007) based on measurements of 5-6 mm aliquot. Note that the pIRIR$_{250}$ *De* is significantly lower than the TT-OSL *De*.**

[Figure]

675

[Figure]

**Figure 4: Bleaching experiments results for TT-OSL (a), VSL (b), and pIRIR (c) signals. Each data point is an average of 3 aliquots (4 for VSL). The TT-OSL signal was defined as first 1 s minus the following 4 s for early subtraction and first 1 s minus the last 5 s for late subtraction.**

[Figure]

680

Figure 5: Dose recovery experim results for the TT-OSL (a) and pIRIR (b) signals. Each data point is an average of 3 or 4 aliquots. The solid lines are 1:1 ratio ±10% (dashed lines).

[Figure]

**Figure 6: Natural saturation profiles of OSL, TT-OSL, VSL, and pIRIR250 signals. The natural luminescence signals of samples are plotted against their depth. Each data point is an average with standard deviation of 4 aliquots. OSL and TT-OSL data is modifieds after Faershtein et al. (2019). The dashed lines are saturation depths of the signals.**

[Figure]

**Figure 7: Semi-natural DRCs of OSL, TT-OSL, and pIRIR₂₅₀.**The natural  signals of samples are plotted against their laboratory measured equivalent doses. The *Ln/Tn* values are average of four aliquots with standard deviation. OSL and TT-OSL data is modifies after Faershtein et al. (2019). Sample below the saturation depth (2 m for OSL and 6 m for TT-OSL and pIRIR₂₅₀) are in grey.

[Figure]

**Figure 8:** TT-OSL ages and pIRIR$_{250}$ uncorrected and corrected (Huntley and Lamothe, 2001; Kars et al., 2008) ages. All ages bellow 6 m should be treated as minimum ages. Some of the pIRIR$_{250}$ corrected ages after Kars et al. (2008) are indicated as $2D_0$ ages (of the natural simulated DRC); therefore, are presented as minimum ages. On the right, quartz environmental dose rates are presented. Note that the TT-OSL ages below 6 m mirror the dose rate pattern.

[Figure]

**Figure 9: Results of the fading correction after Kars et al. (2008) for sample KR-1. Measured, unfaded, and fading corrected (simulated natural) DRCs are presented. For this sample the *Ln/Tn* is above the saturation level of the natural simulated DRC. Inset – fading rates measurement results (following Auclair et al., 2003) for this sample: *g*-value= 1.17±0.36 (% per decade) and $\rho'$=1.32±0.38 (*10⁻⁶).**

700

[Figure]

[Figure]

705  **Figure 10: KR combined luminescence** **chronology. Ages above 6 m are based on uncorrected pIRIR$_{250}$ ages, while ages below 6 m are the oldest of TT-OSL or Kars et al. (2008) corrected pIRIR$_{250}$ ages. The ages of all samples below 6 m are minimum ages.**

[Figure]

[Figure]

**Figure 11: VSL ages, obtained by projecting the *Ln/Tn* values of the samples on the MAAD DRC from a modern sample (DF-13), plotted against TT-OSL SAR ages. Sample below the saturation depth (6 m) are in grey. The dashed line is the 1:1 ratio.**